# Pass@k Training for Adaptively Balancing Exploration and Exploitation of Large Reasoning Models

## Abstract

Reinforcement learning with verifiable rewards (RLVR) typically adopts *Pass@1 as the reward*, causing policies to prefer conservative and similar actions. The RLVR algorithm has faced issues in balancing exploration and exploitation. Thus, identifying an appropriate reward metric is crucial. Motivated by adapting the Pass@k metric as the reward to enhance the model's Pass@k score from previous work, we wonder whether this approach can be used to balance the conflict between the model's exploration and exploitation abilities. To investigate this, we first adopt Pass@k as the reward in the popular RLVR algorithm to train a policy (*i.e.,* **Pass@k Training**) in three variants. During analysis, we observe the improvement in its exploration ability without compromising its exploitation ability. Based on this, our further analysis reveals that exploration and exploitation are not inherently conflicting objectives, while they can mutually enhance each other. However, employing Pass@k as the reward requires complex derivation, making it difficult to be integrated into other algorithms. To alleviate this issue, inspired by the advantage function design process in Pass@k Training, we explore the *implicit reward design* in RLVR, yielding promising results and highlighting a potential future research direction.

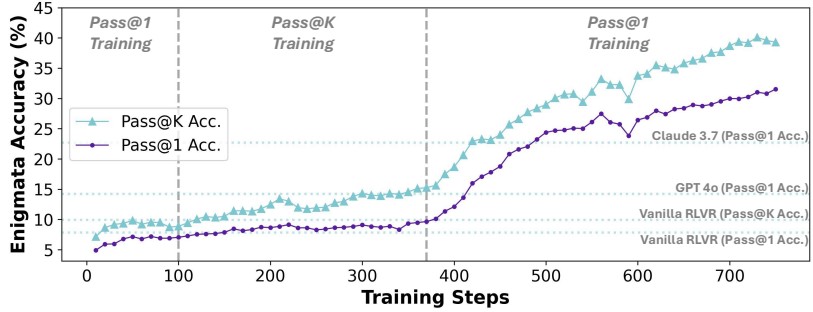

Figure 1: The performance of Qwen2.5-7B-Instruct on Enigmata task (Validation Set) (Chen et al., 2025a), which is a verifiable puzzle task (Appendix B.1). **Pass@k Training** activates the model's exploration ability, surpassing vanilla RLVR and powerful LLMs, *e.g.,* GPT 4o and Claude 3.7.

## 1 Introduction

Recently, reinforcement learning with verifiable rewards (RLVR) has dramatically boosted the reasoning ability of large language models (LLMs) (Trung et al., 2024; Lambert et al., 2024; Zhang et al., 2024). During RLVR, LLMs generate multiple responses based on the given prompt and learn from the external outcome-based assessments (Shao et al., 2024; Kimi et al., 2025; Guo et al., 2025). In such process, LLMs can possess the ability to perform complex reasoning actions (Min et al., 2024), thereby achieving higher performance (DeepSeek-AI et al., 2025; Chen et al., 2025b). The success of the large reasoning models (LRMs), *e.g.,* OpenAI o1 (Jaech et al., 2024) and DeepSeek R1 (DeepSeek-AI et al., 2025), suggests that RLVR pushes the limits of LLM abilities.

The current RLVR training typically optimizes the Pass@1 objective, also known as *Pass@1 Training* (Xie et al., 2025; Hu et al., 2025a), leading to a major challenge of the balance of exploration and exploitation (Cheng et al., 2025; Hou et al., 2025). Typically, exploration refers to performing novel and various actions (Wang et al., 2024), while exploitation requires LLMs to invoke reliable actions that the verifier prefers among the known behaviours (Singh et al., 2024). During the Pass@1 Training, LLMs tend to imitate the behaviours that can bring an increase of reward scores in previous attempts, and prevent the behaviours that receive low rewards (Lyu et al., 2025; Chen et al., 2024). In this case, a solution, which contains sound reasoning paths but yields an incorrect final answer, will still be assigned negative rewards (Chen et al., 2024; Sun et al., 2025). In such cases, these sound reasoning paths are also penalized, which in turn discourages the model from employing similar approaches in subsequent reasoning, thereby diminishing its exploration ability. (Cui et al., 2025) Limited by the suboptimal nature of the reward under RLVR (Nesterov & Spokoiny, 2017; Ghadimi & Lan, 2013), LLMs prefer conservative actions (Li et al., 2021), restricting the advancement potential of scaling the training step.

To mitigate this issue, we advocate for a new RLVR approach with a higher tolerance for incorrect responses, as they might contain useful ideas or behaviors, thereby extending LLM ability boundaries. Fortunately, Pass@k has emerged to assess whether models can generate correct responses within $k$ attempts, assessing LLM ability boundaries (Brown et al., 2024). In previous work (Walder & Karkhanis, 2025; Tang et al., 2025), the Pass@k metric is utilized as the reward function in the reinforcement learning process to enhance the Pass@k performance of LLMs. Compared with the Pass@1 metric, the Pass@k metric allows the policy to generate several incorrect responses. To maximize the probability of generating at least one positive sample, a "smart" policy will generate $k$ solutions with high diversity to cover the possible solution space, rather than $k$ similar samples. Based on the above discussion, we ask a more practical and impactful question: how can Pass@k Training be leveraged to balance the exploration and exploitation ability of LLMs to boost a model's Pass@1 score and generalize to the more complex scenarios (*e.g.,* combining with different RLVR algorithms or situations where reward design is difficult)?

Building on this, we integrate the Pass@k metric as the reward in the popular RLVR algorithm (*i.e.,* DAPO (Yu et al., 2025)) to train a model that has already undergone Pass@1 Training in three variants, *i.e.,* **Pass@k Training**. We find that the trained model can achieve higher Pass@k scores and comparable Pass@1 scores with the vanilla RLVR. To further understand its inner mechanism, we propose five research questions to investigate how Pass@k Training balances the exploration and exploitation of LLMs and what benefits it brings, observing the enhancement of the model's Pass@1 score, which is a key metric for determining real-world deployability. Moreover, to better employ Pass@k Training in various scenarios, we explore how to enhance the effectiveness of RLVR via *implicit reward design* to avoid the complex analytical derivation, observing the strong improvement in both LLM Pass@1 and Pass@k scores. Overall, the vital takeaways can be summarized as follows:

• Compared to Pass@1 Training, **Pass@k Training** significantly enhances the exploration ability of LLMs, improving Pass@k performance while maintaining and even enhancing Pass@1 scores. Among its three variants, bootstrap sampling offers higher training efficiency than full sampling, and analytical derivation mitigates the variance introduced by sampling. (Sec. 2)

• The enhancement of LLM exploration ability is beneficial to their exploitation through further Pass@1 Training, leading 7B LLM to surpass the powerful LLMs, highlighting its practical value. Compared to the variants of Pass@1 Training, Pass@k Training is more robust and effective. (Sec. 3)

• Pass@k Training with analytical derivation, which directly designs the advantage function, can be viewed as a form of *implicit reward design*. Following this idea, empirical experiments suggest that implicit reward design allows finer-grained control over optimization, without complex theoretical derivations, making it a promising direction for future RLVR development. (Sec. 4)

## 2 PASS@K AS REWARD IN RLVR TRAINING

In this section, we first formulate the reasoning tasks and Pass@1 Training (Sec. 2.1). Next, we introduce how to implement Pass@k as a reward in the RLVR process (Sec. 2.2), and then propose two enhancements (Sec. 2.3 and Sec. 2.4). We present a demonstration in Fig. 2, and provide the implementation details of the experiment settings and benchmarks in Appendix B.

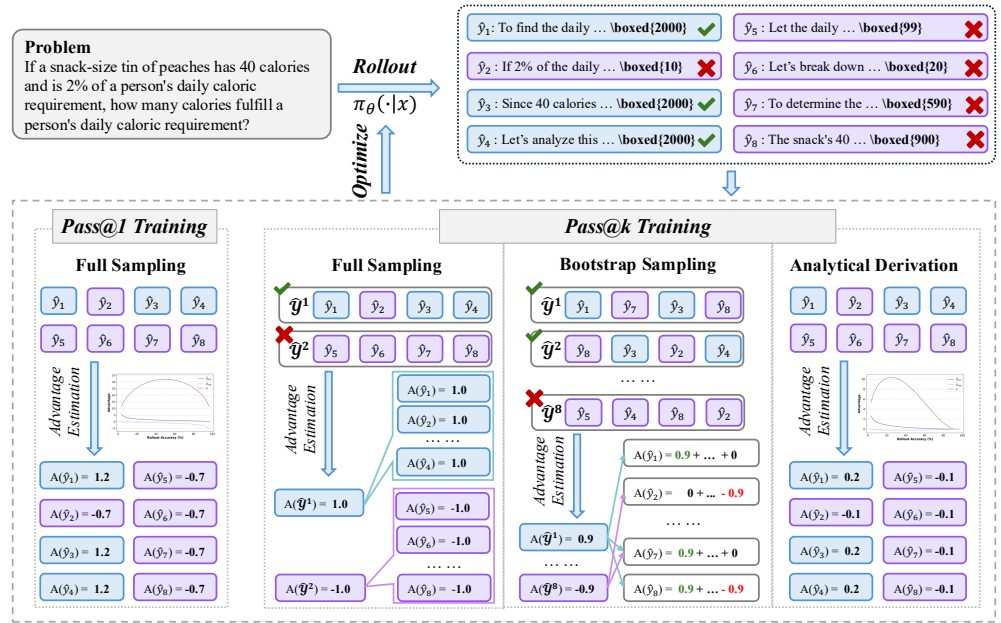

Figure 2: The overview and comparison between Pass@1 and Pass@k Training. Full sampling, bootstrap sampling, and analytical derivation are three progressive enhancements for the Pass@k Training. We present the pseudo-code in Appendix D to better demonstrate them.

## 2.1 FORMULATION OF REASONING TASKS AND PASS@1 TRAINING

Typically, a reasoning problem from the whole dataset $D$ contains a question description $x$ and an answer $y$. The policy $\pi_\theta$ (*i.e.,* LLM with the parameters $\theta$) needs to generate a response $\hat{y} = \{t_1, \ldots, t_l\}$ based on $x$, where $t_i$ and $l$ refer to the $i$-th token and the response length $\hat{y}$. A verifier verifies the correctness of the LLM response and provides a reward $R(y, \hat{y}) \in \{R_{\text{neg}}, R_{\text{pos}}\}$, where $R_{\text{neg}}$ and $R_{\text{pos}}$ are for negative and positive responses, respectively. Simplify, we use $R$ to represent $R(y, \hat{y})$. In practice, we adopt $R_{\text{neg}} = 0$ and $R_{\text{pos}} = 1$.

Based on the above formulation of reasoning tasks, in the Pass@1 Training process (*e.g.,* GRPO (Shao et al., 2024)), the advantage is estimated through the average value and standard deviation of the response rewards within the same group, which can be shown as follows,

$$\bar{R} = \frac{1}{N_{\text{rollout}}} \sum_{i=1}^{N_{\text{rollout}}} R_i, \ \ \sigma = \frac{1}{N_{\text{rollout}}} \sqrt{\sum_{i=1}^{N_{\text{rollout}}} (R_i - \bar{R})^2}, \ \ \hat{A}_{i,1} = \cdots = \hat{A}_{i,|\hat{y}_i|} = \frac{R_i - \bar{R}}{\sigma}, \quad (1)$$

where $N_{\text{rollout}}$ denotes the number of the rolled-out responses, and $R_i$ and $\hat{y}_i$ refer to the rewards and the content of the $i$-th response, respectively. After obtaining the advantage, GRPO utilizes the following equation to calculate the objective function $\mathcal{J}(\theta)$ that is leveraged to optimize the policy,

$$\mathcal{J}(\theta) = \mathbb{E}_{(q,a) \sim D, \{o_i\}_{i=1}^G \sim \pi_\theta(\cdot|q)} \left[ \frac{1}{G} \sum_{i=1}^G \frac{1}{|\hat{y}_i|} \sum_{t=1}^{|\hat{y}_i|} \left( \min \left( r_{i,t} \hat{A}_{i,t}, \text{clip} \left( r_{i,t}, 1 - \varepsilon, 1 + \varepsilon \right) \hat{A}_{i,t} \right) - \beta D_{\text{KL}} \right) \right].$$

$$(2)$$

Since each token shares the same advantage value in GRPO, we will no longer distinguish at the token level in the following, and use $\hat{A}_i$ to represent the advantage of the $i$-th response, instead.

## 2.2 SIMPLE PASS@K TRAINING VIA FULL SAMPLING

The behaviour of LLMs can be adjusted by the corresponding rewards (Tang et al., 2025). Motivated by this, since the Pass@k metric can reflect LLM exploration ability, we first introduce the definition of the Pass@k metric and then incorporate it into the RLVR reward function.

**Definition of Pass@k Metric.** Given the question $x$, the policy model is utilized to rollout the $k$ responses through a specific decoding strategy (*e.g.,* sampling-based decoding strategy or Monte Carlo Tree Search). The $i$-th sampled response $\hat{y}_i$ will receive a reward $R_i$, which is provided by the verifier. Based on this, the value of the Pass@k metric is defined as the expected maximum reward obtained from these $k$ responses. Formally, the Pass@k metric can be computed as follows,

$$\text{Pass@k} = \mathbb{E}_{(x,y) \sim D, \{\hat{y}_i\}_{i=1}^k \sim \pi_\theta(\cdot|x)} \left[ \max (R_1, \ldots, R_k) \right]. \tag{3}$$

**Full Sampling.** First, we rollout the responses $\hat{\mathcal{Y}} = \{\hat{y}_1, \ldots, \hat{y}_{N_\text{rollout}}\}$ from $\pi_\theta$ for the given question and separate these responses into $N^\text{group} = \lfloor \frac{N_\text{rollout}}{k} \rfloor$ groups (discarding the redundancy), where the $j$-th group contains the $k$ responses $\hat{\mathcal{Y}}^j = \{\hat{y}_{k \times (j-1)+1}, \ldots, \hat{y}_{k \times (j-1)+k}\}$. Next, the group reward is computed by the maximum reward of the responses it contains. Following the GRPO advantage estimation approach, the advantage of $j$-th group $\hat{A}^j$ can be calculated. After that, we divide the group advantage to the responses it contains, *i.e.,* $\hat{A}_{k \times (j-1)+1} = \cdots = \hat{A}_{k \times (j-1)+k} = \hat{A}^j$. Finally, we can utilize the sampled responses and their advantage value to optimize the policy.

**Empirical Insight: Improving Exploration.** We compare the performance between Pass@k Training with full sampling and the Pass@1 Training in Fig. 7 and Table 1. We observe that the model's Pass@k performance is slightly improved during the Pass@1 Training. As a result, while the Pass@1 metric improves in the early stages of training, it stagnates in the later stages, indicating that the model has fallen into a local optimum. In contrast, employing Pass@k as the reward, the Pass@k scores of LLMs achieve a larger improvement, showing that the exploration ability of LLMs can be enhanced. Moreover, Pass@k Training results in Pass@1 score gains, showing the similarity of their training objective. However, the improvement in Pass@1 scores is not as significant as that of Pass@1 Training.

Table 1: Performance on Maze task of Pass@1 Training and Pass@k Training with Full Sampling (FS).

| Method | Pass@1 | Pass@k |
|---|---|---|
| Backbone | 18.3 | 25.0 |
| Pass@1 Training | **32.4** | 33.0 |
| FS w/ $N_\text{rollout} = 32$ | 26.4 | **47.6** |

**Takeaway from Sec. 2.2.** Compared to conventional RLVR that employs Pass@1 as the reward function, adopting Pass@k as the reward not only leads to substantial improvements in the model's Pass@k performance on downstream tasks but also yields notable gains in Pass@1 scores.

## 2.3 EFFICIENT PASS@K TRAINING VIA BOOTSTRAP SAMPLING

Although Pass@k Training with Full Sampling can improve LLM exploration ability, the rollout times increase significantly with the increase in the number of groups, costing more resources. Besides, we think that the noise introduced during the Full Sampling process may affect the training effect of Pass@1. Thus, we utilize the bootstrap sampling to reduce the rollout times while maintaining the number of groups, and alleviate the influence of noisy rewards.

**Bootstrap Sampling.** We first rollout the responses $\hat{\mathcal{Y}} = \{\hat{y}_1, \ldots, \hat{y}_{N_\text{rollout}}\}$ from the policy $\pi_\theta$. To construct $N^\text{group}$ groups for Pass@k Training, we randomly sample $k$ responses from $\hat{\mathcal{Y}}$, and these sampled responses constitute a group. Concretely, to construct the $j$-th group, we select $k$ distinct indices of the generated responses $\mathcal{P} = \{p_{j,1}, \ldots, p_{j,k}\}$. The selected responses constitute the $j$-th group $\hat{\mathcal{Y}}^j = \{\hat{y}_{p_{j,1}}, \ldots, \hat{y}_{p_{j,k}}\}$. We repeat this process $N^\text{group}$ times to collect $N^\text{group}$ groups. Next, we estimate the advantage for each group and assign it to the responses. Since we use a bootstrap sampling to construct groups, some responses may appear in multiple groups.

Table 2: Performance on Maze task of Pass@1 Training and Pass@k Training with Bootstrap Sampling (BS).

| Method | Pass@1 | Pass@k |
|---|---|---|
| Pass@1 Training | 32.4 | 33.0 |
| FS w/ $N_\text{rollout} = 128$ | 28.9 | **53.1** |
| FS w/ $N_\text{rollout} = 32$ | 26.4 | 47.6 |
| BS w/ $N_\text{rollout} = 32$ | **37.8** | 51.0 |

We compute the advantage of each response by summing the advantages of the groups it belongs to,

$$\hat{A}_i = \sum_{j=1}^{N^\text{group}} \hat{A}^j \cdot \mathbb{I}[\hat{y}_i \in \hat{\mathcal{Y}}^j], \tag{4}$$

where $\mathbb{I}[\hat{y}_i \in \hat{\mathcal{Y}}^j]$ is an indicator function that returns 1 if and only if the $i$-th response $\hat{y}_i$ belongs to the $j$-th group $\hat{\mathcal{Y}}^j$, while returns 0 for others. In practice, we set $N^{\text{group}} = N_{\text{rollout}}$.

**Empirical Insight: Reducing Training Budget.** In Fig. 8 and Table 2, given the same rollout number $N_{\text{rollout}}$ (*i.e.,* "$N_{\text{rollout}} = 32$ w/ Full Sampling" v.s. "$N_{\text{rollout}} = 32$ w/ Bootstrap Sampling"), bootstrap sampling outperforms full sampling. This improvement arises from the fact that bootstrap sampling generates a larger number of groups, reducing the variance of the sampled reward distribution to the true distribution, leading to more stable training. With the same number of groups $N^{\text{group}}$, bootstrap sampling does not cause significant performance degradation compared to "$N_{\text{rollout}} = 128$ w/ Full Sampling", and it requires one-fourth of the computational cost, showing higher efficiency. Additionally, it achieves comparable performance to Pass@1 Training on the Pass@1 scores.

**Takeaway from Sec. 2.3.** Compared with the full sampling implementation of Pass@k Training, the bootstrap sampling method can achieve better results with the same rollout number. With the same number of groups, it can reduce computational overhead and achieve comparable performance.

## 2.4 ANALYTICAL DERIVATION FOR EFFICIENT AND EFFECTIVE PASS@K TRAINING

Following the idea of the bootstrap sampling (Sec. 2.3), we derive the analytical solution of the response advantage (*i.e.,* $\hat{A}_{pos}$ and $\hat{A}_{neg}$) to remove the variance brought by the sampling operation for constructing the groups. The details of the derivation are presented in Appendix C.

**Analytical Derivation.** A group that contains at least one positive response (*positive group*) will be assigned the positive reward $R_{\text{pos}}$, while the other groups (*negative group*) will be endowed with the negative reward $R_{\text{neg}}$. We start by analyzing the average reward and standard deviation of the groups, *i.e.,* $\bar{R}^{\text{group}}$ and $\sigma^{\text{group}}$. First, the average reward of the group can be formulated:

$$\bar{R}^{\text{group}} = \frac{1}{N_{\text{total}}^{\text{group}}} \times \left( N_{\text{pos}}^{\text{group}} \times R_{\text{pos}} + N_{\text{neg}}^{\text{group}} \times R_{\text{neg}} \right), \tag{5}$$

where $N_{\text{total}}^{\text{group}}$ refers to the total number of groups, and $N_{\text{pos}}^{\text{group}}$ and $N_{\text{neg}}^{\text{group}}$ denote the number of positive and negative groups, respectively. To compute them, we first define the number of positive and negative responses as $N_{\text{pos}}$ and $N_{\text{neg}}$, respectively. As each group is constructed by selecting $k$ responses, we can obtain the total number of the group $N_{\text{total}}^{\text{group}} = \binom{N_{\text{rollout}}}{k}$. Since negative groups do not contain the positive responses, when and only when randomly sampling $k$ negative responses from $\hat{\mathcal{Y}}$, these sampled responses can construct a negative group. Thus, the number of negative and positive groups can be calculated, *i.e.,* $N_{\text{neg}}^{\text{group}} = \binom{N_{\text{neg}}}{k}$ and $N_{\text{pos}}^{\text{group}} = \binom{N_{\text{rollout}}}{k} - \binom{N_{\text{neg}}}{k}$. Based on above derivation, we can obtain $\bar{R}^{\text{group}}$, and then deduce the standard deviation $\sigma^{\text{group}}$,

$$\bar{R}^{\text{group}} = 1 - \frac{\binom{N_{\text{neg}}}{k}}{\binom{N_{\text{rollout}}}{k}}, \quad \sigma^{\text{group}} = \sqrt{\bar{R}^{\text{group}} \times \left(1 - \bar{R}^{\text{group}}\right)}. \tag{6}$$

Based on the average reward $\bar{R}^{\text{group}}$ and the standard deviation $\sigma^{\text{group}}$ (Eq. 6) of reward scores, we can deduce the advantage of the positive and negative group (*i.e.,* $\hat{A}_{\text{pos}}^{\text{group}}$ and $\hat{A}_{\text{neg}}^{\text{group}}$) as follows,

$$\hat{A}_{\text{pos}}^{\text{group}} = \frac{R_{\text{pos}} - \bar{R}^{\text{group}}}{\sigma^{\text{group}}} = \frac{1 - \bar{R}^{\text{group}}}{\sigma^{\text{group}}}, \quad \hat{A}_{\text{neg}}^{\text{group}} = \frac{R_{\text{neg}} - \bar{R}^{\text{group}}}{\sigma^{\text{group}}} = -\frac{\bar{R}^{\text{group}}}{\sigma^{\text{group}}}. \tag{7}$$

To compute the response advantage $\hat{A}_{pos}$ and $\hat{A}_{neg}$, we need to consider the correctness of the group each response belongs to. Typically, a response will belong to $\binom{N_{\text{rollout}}-1}{k-1}$ groups, because a group can be formed with the current response if and only if $k-1$ responses are selected from the remaining $N_{\text{rollout}} - 1$ responses. Further, for a positive response, the groups that it belongs is always the positive group. Thus, *the advantage of a positive response $\hat{A}_{pos}$ can be calculated as follows,*

$$\hat{A}_{\text{pos}} = \left(1 - \bar{R}^{\text{group}}\right) \times \left(\sigma^{\text{group}}\right)^{-1}. \tag{8}$$

For a negative response, it belongs to a negative group if and only if with other $k-1$ negative responses. Thus, the number of such groups is $\binom{N_{\text{neg}}-1}{k-1}$. Based on this, we can compute the number

of positive groups by subtracting the number of negative groups from the total, *i.e.*, $\binom{N_{\text{rollout}}-1}{k-1} - \binom{N_{\text{neg}}-1}{k-1}$. Thus, *the advantage of a negative response $\hat{A}_{neg}$ can be calculated as follows,*

$$\hat{A}_{\text{neg}} = \left( 1 - \bar{R}^{\text{group}} - \frac{\binom{N_{\text{neg}}-1}{k-1}}{\binom{N_{\text{rollout}}-1}{k-1}} \right) \times \left( \sigma^{\text{group}} \right)^{-1}. \qquad (9)$$

After obtaining the analytical solutions of response-relative advantage $\hat{A}_{\text{pos}}$ and $\hat{A}_{\text{neg}}$, we directly employ them in the advantage estimation process and then optimize the model parameters.

**Discussion of Response Advantage $\hat{A}_{\text{pos}}$ and $\hat{A}_{\text{neg}}$.** First, during Pass@k Training, $\hat{A}_{\text{pos}} \geq 0$ and $\hat{A}_{\text{neg}} \leq 0$ consistently, reinforcing positive responses and penalizing negative ones. Analytical Derivation further improves reward accuracy by eliminating noise. Second, compared to Pass@1 Training, Pass@k Training emphasizes difficult problems while downweighting easier ones, likely boosting the model's exploration ability. Detailed analysis is provided in Appendix G. Third, reviewing the Pass@k Training, we find that directly modifying the advantage function can improve the corresponding metrics. The explicit design of advantage can be regarded as the **implicit reward design**. In this method, we do not focus on the specific form of the reward function; rather, our attention is directed solely toward the training objective. In Sec. 4, we explore this approach and find that it can integrate Pass@k Training with other methods, yielding improved optimization outcomes, highlighting it as a highly promising research direction.

**Empirical Insight: Further Improving Pass@k.** We compare the effects of Pass@1 Training and Pass@k Training with bootstrap sampling and analytical derivation with $N_{\text{rollout}} = 32$, and conduct experiments on different LLMs on various tasks in Appendix F. In Fig. 9 and Table 3, we observe that Pass@k Training achieves higher Pass@k scores than Pass@1 Training, further confirming its effectiveness. When the training steps increase, the bootstrap sampling faces a performance fluctuation, which indicates its instability. In contrast, analytical derivation eliminates the sampling process for group construction, which reduces the variance caused by this process, providing a more stable RLVR.

Table 3: Performance on Maze task of Pass@1 Training and Pass@k Training with Analytical Derivation (AD).

| Method | Pass@1 | Pass@k |
|---|---|---|
| Pass@1 Training | 32.4 | 33.0 |
| FS w/ $N_{\text{rollout}} = 32$ | 26.4 | 47.6 |
| BS w/ $N_{\text{rollout}} = 32$ | 37.8 | 51.0 |
| AD w/ $N_{\text{rollout}} = 32$ | **94.6** | **100.0** |

Besides, it also enhances the policy's Pass@1 scores. Thus, according to empirical experiments, Pass@k Training with analytical derivation can reduce fluctuations during the training process and bring about continuous performance improvements as the number of training steps increases.

**Takeaway from Sec. 2.4.** Pass@k Training with analytical derivation not only avoids the computational overhead caused by a large number of rollouts in full sampling, but also eliminates the variance introduced by the sampling process, making the RLVR process more efficient and effective, and guiding the model's exploration ability to continuously improve when training steps scale.

## 3 BALANCING EXPLORATION AND EXPLOITATION WITH PASS@K TRAINING

First, to understand how Pass@k Training affects LLM exploration ability, we compare Pass@k Training with variants of Pass@1 Training (Sec. 3.1) and examine the diversity of model responses and the entropy of policy distribution (Sec. 3.2). Moreover, as RLVR robustness is widely concerned (He et al., 2025), we analyze the influence of value $k$ (Sec. 3.3) and the backbone RLVR algorithm (Sec. 3.4). Finally, we explore how to transfer the benefits to Pass@1 scores of LLMs (Sec. 3.5). Further discussions are in Appendix H.

### 3.1 HOW DOES PASS@K TRAINING COMPARE TO THE VARIANTS OF PASS@1 TRAINING?

Motivated by the Pass@k Training and the previous work (Hou et al., 2025), we conduct a comparison between Pass@k Training and two baselines, *i.e.,* Noise Rewards and Entropy Regularization.

**Noise Rewards.** In Sec. 2.2, the negative responses may receive the positive reward if they belong to a positive group. This raises the question of whether the improvement in Pass@k scores is partially driven by learning from these negative responses with positive rewards. To investigate this, we conduct an experiment in which a certain ratio (*i.e.,* 10%, 30%, and 50%) of negative responses are assigned a positive reward. In Fig. 3a, guiding LLMs to learn from

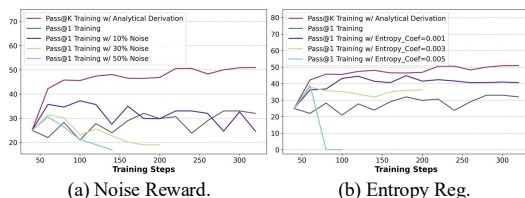

(a) Noise Reward.    (b) Entropy Reg.

Figure 3: Pass@k scores of Maze tasks in RLVR.

negative responses does not contribute to their reasoning ability, and a higher ratio of noise or additional training steps even degrades model ability, suggesting that naively incorporating noise into the reward cannot enhance exploration ability.

**Entropy Regularization.** A surge of studies (Cui et al., 2025; Wang et al., 2025) have found that entropy can indicate the exploration ability of LLMs and can be incorporated into RLVR to preserve their ability. Following this idea, we employ entropy regularization with the coefficient of $\{0.001, 0.003, 0.005\}$ in the RLVR process (Fig. 3b). We can find that a high coefficient might cause the collapse of the model, *e.g.,* "Coef=0.005". Although the small coefficient does not make LLM crush, it still underperforms Pass@k Training, and even leads to a decrease in the performance of LLMs when the training steps increase, indicating that entropy regularization might affect training effectiveness and stability. Moreover, we discuss other entropy-guided methods in Appendix H.3.

**Takeaway from Sec. 3.1.** Pass@k Training outperforms Noise Rewards and Entropy Regularization: randomly flipping the reward of negative responses might degrade the model's ability, and integrating Entropy Regularization brings new trade-off issues that hardly achieves improvement.

## 3.2 DOES PASS@K TRAINING REALLY IMPROVE THE EXPLORATION ABILITY OF LLMS?

To analyze the changes in exploration of LLMs during the RLVR process, we conduct the related empirical study from the perspective of answer diversity and entropy of policy distribution.

**Answer Diversity of Negative Responses.** We counter the accuracy and the ratio of different answers among the negative responses of Pass@k and Pass@1 Training, aiming to assess the exploration ability of LLMs on the uncertain answer. In Fig. 4a, we observe that the answer diversity of the negative response stays at the same level during the Pass@1 training, indicating that the LLMs try to select the "safe" actions and tend to generate similar answers, limiting the scope of exploration and the effect of

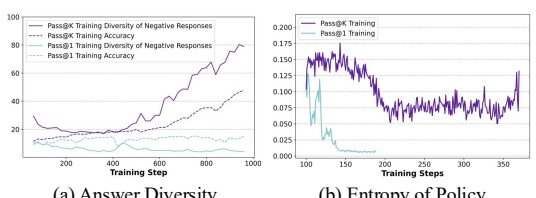

(a) Answer Diversity.    (b) Entropy of Policy.

Figure 4: Results on Maze task in RLVR.

RLVR. Differently, in Pass@k Training, the model is encouraged to achieve a higher Pass@k score and naturally learn the strategy to generate diverse answers when the model is not confident enough. Thus, the exploration ability of LLM is enhanced and thereby improves its exploitation ability.

**Entropy of Policy Distribution.** In Fig. 4b, Pass@k Training keeps the entropy of policy distribution at a relatively high level, while Pass@1 Training induces entropy to converge to a low value, suggesting that LLMs can keep their exploration ability during Pass@k Training, but will lose it during Pass@1 Training. Besides, we also observe that Pass@k Training leads to an increase in entropy, starting from around 200 steps, validating our hypothesis that using Pass@k as the training objective can encourage the model to conduct more exploration, thereby naturally increasing entropy.

**Takeaway from Sec. 3.2.** Pass@k Training guide LLMs to conduct more exploration, *i.e.,* generating diverse answers, which naturally increase entropy, when the model is not confident enough to generate the correct answer, suggesting that exploration and exploitation can be improved mutually.

Table 4: Pass@k performance of Enigmata of different RLVR methods. "P@k T." denotes the Pass@k Training with analytical derivation.

| | Crypto | Arithmetic | Logic | Grid | Graph | Search | Sequential | Overall |
|---|---|---|---|---|---|---|---|---|
| Qwen2.5-7B-Ins | 0.7 | 3.3 | 48.4 | 9.2 | 11.6 | 1.0 | 5.4 | 10.1 |
| + DAPO | 5.7 | 28.0 | 68.4 | 22.8 | 20.4 | 6.8 | 12.7 | 21.3 |
| + DAPO & P@k T. | 39.7 | 63.0 | 74.0 | 27.8 | 21.5 | 12.3 | 18.3 | **29.8 (+8.5)** |
| + GRPO | 0.3 | 13.0 | 65.6 | 19.4 | 14.4 | 2.6 | 12.4 | 17.3 |
| + GRPO & P@k T. | 0.7 | 17.7 | 64.2 | 20.8 | 18.2 | 4.0 | 12.7 | **18.6 (+1.3)** |
| + ForkingToken | 94.0 | 35.0 | 67.1 | 25.9 | 21.6 | 13.4 | 11.7 | 29.1 |
| + ForkingToken & P@k T. | 91.3 | 48.3 | 67.6 | 27.4 | 27.8 | 22.3 | 18.6 | **34.0 (+4.9)** |

### 3.3 HOW DOES THE VALUE OF κ AFFECT PASS@K TRAINING?

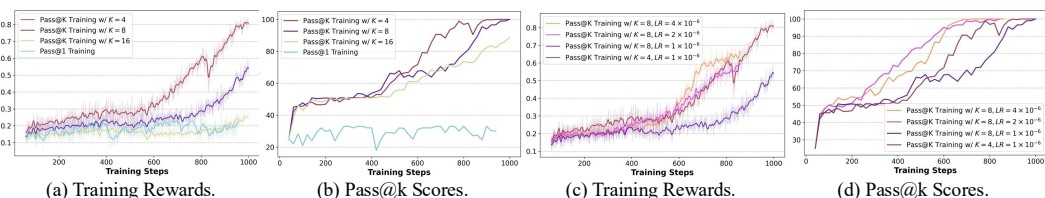

(a) Training Rewards.  (b) Pass@k Scores.  (c) Training Rewards.  (d) Pass@k Scores.

Figure 5: Training progress on Maze Tasks of Pass@k Training under various $k$ and learning rate (LR). Specifically, sub-figures (a) and (b) are in the setting of LR $= 1 \times 10^{-6}$.

We adjust $k$ in $\{4, 8, 16\}$ in the RLVR (Fig. 5a and Fig. 5b). For various $k$, the training rewards and test set Pass@k scores can be improved to a relatively high level, indicating the value of $k$ is not a vital factor that improves the exploration ability of LLMs. However, the improvement slows down with the increase of $k$. Through analyzing the analytical solutions (*i.e.,* Eq. 8 and Eq. 12), a larger value of $k$ brings a smaller value of advantage, resulting to a shorter optimization step and lower training efficiency. To address this, we investigate whether enlarging the learning rate (LR) can improve training efficiency. Following this idea, we employ the learning rate in $\{1 \times 10^{-6}, 2 \times 10^{-6}, 4 \times 10^{-6}\}$ on the setting of $N = 32$ and $k = 8$. In Fig. 5c and Fig. 5d, with the increase of learning rate, the inflection point appears earlier, indicating higher training efficiency. The training efficiency of $k = 8$ even outperforms $k = 4$ when we employ $4 \times 10^{-6}$ as the learning rate. These results have shown that the issues of training efficiency can be easily mitigated.

**Takeaway from Sec. 3.3.** Pass@k Training exhibits strong robustness to the choice of the value of $k$, leading to a stable and effective training process. Although there is a decline in the model's optimization efficiency as $k$ increases, this issue can be addressed by enlarging the learning rate.

### 3.4 CAN PASS@K TRAINING BE INTEGRATED INTO DIFFERENT RLVR ALGORITHMS?

A key open question is whether Pass@k Training can be integrated into other reinforcement learning algorithms. This determines its practicality for real-world deployment and its ability to generalize across diverse application scenarios. To assess the adaptability of Pass@k Training, we combine it with multiple popular RLVR algorithms and test whether Pass@k Training can enhance the model's exploration ability when built upon different backbone algorithms, including DAPO (Yu et al., 2025), GRPO (Shao et al., 2024), and ForkingToken (Wang et al., 2025). According to the experiment results in Table 4, we can observe that Pass@k Training can be adapted to different RLVR algorithms, achieving an improvement in the Pass@k score of LLM. This phenomenon indicates that current popular RLVR algorithms can integrate the Pass@k Training to further improve the exploration ability of LLMs.

**Takeaway from Sec. 3.4.** Pass@k Training can be flexibly integrated into various RLVR algorithms, consistently boosting LLMs' Pass@k performance and enhancing their exploration ability.

Table 5: Pass@1/Pass@k performance of Enigmata of different methods. "P@1 T." and "P@k T." denote the Pass@1 Training and Pass@k Training with analytical derivation based on DAPO.

| | Crypto | Arithmetic | Logic | Grid | Graph | Search | Sequential | Overall |
|---|---|---|---|---|---|---|---|---|
| **Closed-source LLMs w/o RLVR Training (Pass@1)** | | | | | | | | |
| GPT-4o-1120 | 26.2 | 1.9 | 34.5 | 17.8 | 19.3 | 6.0 | 3.9 | 14.2 |
| Claude-3.7-Sonnet | 38.1 | 16.7 | 60.0 | 22.9 | 22.4 | 7.8 | 15.0 | 22.7 |
| Qwen2.5-7B-Ins | 0.1/0.7 | 1.0/3.3 | 28.1/48.4 | 3.7/9.2 | 3.0/11.6 | 0.3/1.0 | 2.7/5.4 | 4.7/10.1 |
| + PKPO | 0.0/0.0 | 1.6/8.0 | 32.4/43.6 | 11.5/20.5 | 9.8/12.7 | 3.7/8.1 | 8.9/10.2 | 10.1/15.4 |
| + ForkingToken | 78.9/94.0 | 10.7/35.0 | 46.2/67.1 | 19.6/25.9 | 14.6/21.6 | 7.4/13.4 | 10.0/11.7 | 20.6/29.1 |
| + EntropyAdv | 46.7/73.3 | 31.3/69.3 | 48.6/72.7 | 24.4/31.5 | 17.0/31.5 | 8.8/19.5 | 16.2/18.7 | 23.3/35.8 |
| + P@1 T. | 1.2/5.7 | 6.6/28.0 | 41.1/68.4 | 14.8/22.8 | 14.5/20.4 | 3.3/6.8 | 9.6/12.7 | 12.9/21.3 |
| + P@k T. | 14.0/39.7 | 27.4/63.0 | 46.0/74.0 | 18.8/27.8 | 15.2/21.5 | 5.3/12.3 | 14.1/18.3 | 17.9/29.8 |
| + P@k T. + P@1 T. | 96.9/98.3 | 36.2/67.7 | 49.3/71.8 | 30.9/37.5 | 20.3/30.7 | 25.8/37.5 | 10.6/12.9 | **30.8/40.6** |
| Qwen2.5-32B-Ins | 1.5/5.3 | 4.6/16.0 | 45.7/71.1 | 11.6/20.6 | 8.0/26.7 | 2.3/6.4 | 7.7/16.3 | 10.9/21.6 |
| + P@1 T. | 95.8/99.7 | 53.0/85.0 | 76.6/92.2 | 38.4/47.4 | 44.4/57.8 | 47.0/58.8 | 21.8/25.8 | 45.2/56.0 |
| + P@k T. | 93.8/99.3 | 51.1/86.3 | 74.8/92.4 | 39.0/49.6 | 42.7/61.3 | 45.9/59.9 | 21.5/26.6 | 44.5/57.4 |
| + P@k T. + P@1 T. | 95.9/99.3 | 49.6/84.3 | 82.0/94.9 | 40.0/51.0 | 48.2/60.2 | 48.8/60.8 | 22.2/26.2 | **46.8/57.9** |

Table 6: Pass@1/Pass@k Performance on mathematical tasks of DeepSeek-R1-Distill-Qwen-1.5B models trained on different approaches. "P@1 T." and "P@k T." denote the Pass@1 and Pass@k Training with analytical derivation, respectively. "×2" refers to repeating the process twice.

| | AIME 2024 | AIME 2025 | OlymMATH-Easy | OlymMATH-Hard | Avg. |
|---|---|---|---|---|---|
| Baseline | 22.7/61.4 | 20.5/37.2 | 6.6/36.7 | 0.6/5.2 | 12.6/35.1 |
| + P@1 T. | 36.7/76.0 | 28.8/49.4 | 16.7/51.7 | 2.5/17.5 | 21.2/48.7 |
| + P@k T. | 36.5/79.3 | 27.0/55.5 | 17.6/59.3 | 2.4/17.4 | 20.9/52.9 |
| + P@k T. + P@1 T. | 42.3/71.7 | 30.4/57.8 | 20.7/60.5 | 3.4/18.9 | 24.2/52.2 |
| + (P@k T. + P@1 T.) × 2 | 44.2/77.2 | 31.5/57.7 | 22.6/62.7 | 4.4/21.2 | **25.7/54.7** |

## 3.5 IS THE PASS@K TRAINING BENEFICIAL FOR PASS@1 PERFORMANCE?

To transfer the benefits from Pass@k to Pass@1 scores, a native method is performing Pass@1 Training on the model that has undergone Pass@k Training. We present the results of Puzzle tasks in Table 5 and mathematical tasks in Table 6, and provide the significance test in Appendix F.1. Moreover, we also conduct the experiments with different LLMs on various tasks in Appendix F.

First, the Pass@1 Training following Pass@k Training can significantly improve the reasoning ability of LLMs, achieving remarkable Pass@1 performance. We can observe that even the 7B model can surpass the powerful closed-source LLMs, *e.g.,* GPT-4o, and Claude-3.7-Sonnet. This might be because Pass@k Training activates the exploration ability of the LLM, unleashing its potential in the subsequent RLVR training. Second, either the small-scale or large-scale LLMs (*e.g.,* 7B or 32B model) can benefit from Pass@k Training. Third, the domain of tasks also does not affect the transfer from LLM Pass@k scores to their Pass@1 scores. Our evaluation includes puzzle tasks and mathematical tasks, requiring LLMs to possess different abilities. Pass@k Training can enhance the corresponding abilities, showing strong effectiveness.

Besides, we compare the effectiveness of Pass@k Training with other competitive baseline approaches, including PKPO (Walder & Karkhanis, 2025), ForkingToken (Wang et al., 2025), and EntropyAdv (Cheng et al., 2025), which are proposed to enhance the Pass@k performance of LLMs. Based on the evaluation results, Pass@k Training (*i.e.,* "+ P@k T. + P@1 T.") outperforms these baselines in both Pass@1 and Pass@k scores, showing its effectiveness. Moreover, the Pass@1 score of Pass@k Training is large higher than these baselines, demonstrating the practical value of Pass@k Training for real-world applications.

**Takeaway from Sec. 3.5.** The benefits brought by Pass@k Training can be transferred to Pass@1 performance of LLMs, which is not affected by the scale of model parameters (*e.g.,*, 7B or 32B), or the domain of downstream tasks (puzzle or mathematical tasks).

# 4 GENERALIZING PASS@K TRAINING VIA IMPLICIT REWARD DESIGN

From Sec. 2.4, employing Pass@k Training to new scenarios needs a complex derivation, harming its practical potential. To mitigate it, we explore how to design the advantage function for optimization objectives, *i.e., Implicit Reward Design*. We consider how to combine the strengths of Pass@1 and Pass@k Training as the experimental scenario, since its analytical solutions are hard to derive.

## 4.1 IMPLICIT REWARD DESIGN

**Combination based on Guidance of Accuracy: Combination Training.** We consider whether combining Pass@1 and Pass@k Training could be beneficial. Thus, we design the following formula to estimate the final advantage value: $\hat{A} = \frac{N_{\text{pos}}}{N} \times \hat{A}_{\text{Pass@k}} + (1 - \frac{N_{\text{pos}}}{N}) \times \hat{A}_{\text{Pass@1}}$, where $\hat{A}_{\text{Pass@k}}$ and $\hat{A}_{\text{Pass@k}}$ denote the advantage values estimated by Pass@k and Pass@1 Training approach, respectively. In the above equation, when the sampled response has a low accuracy, the advantage value from Pass@1 Training receives a higher weight and dominates the training process, leading to high training efficiency. Conversely, when the accuracy is high, the advantage value of Pass@k Training is assigned a greater weight, avoiding LLMs from overfitting to the easy problems.

**Adaptive Training based on Policy Entropy: Adaptive Training.** The entropy of policy distribution can indicate its exploration ability (Cui et al., 2025). Thus, we conduct the Pass@k Training based on the guidance of policy entropy. Concretely, we first compute the average entropy $\bar{E}$ of the sampled responses of each question, and then rank each problem based on its $\bar{E}$. We designate the top 50% as *high-exploration problems* and the rest as *low-exploration problems*. For high-exploration problems, we use the Pass@1 advantage function to help the model exploit prior exploration, and apply the Pass@K advantage function to encourage further exploration for others.

## 4.2 ANALYSIS AND DISCUSSION

We present the results of Qwen on Enigmata in Fig. 6, and observe that the variants outperform the standard Pass@k Training. Both Pass@1 and Pass@k scores improve rapidly and maintain a high growth rate during RLVR, indicating the complementarity of Pass@1 and Pass@K Training. By designing a proper adaptation mechanism, it is possible to better leverage the strengths of these methods, enabling the model to achieve higher performance. In contrast, Pass@k Training leads to slower perfor-

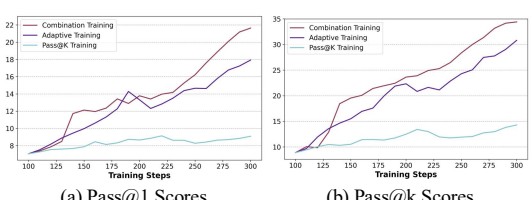

(a) Pass@1 Scores.          (b) Pass@k Scores.

Figure 6: Pass@k Training and variants in RLVR.

mance gains. This is because Pass@k Training encourages LLMs to perform exploration, leading to a slow convergence rate. The analysis supports the idea that designing an advantage function based on the model's current state can effectively enhance its abilities. The idea of implicit reward design is an extension of Pass@k Training, and it also provides a solution for scenarios where reward function design is challenging, highlighting a potential future direction.

# 5 CONCLUSION

In this work, we proposed the **Pass@k Training**, aiming to enable mutual improvement between the exploration and exploitation abilities of the LLM, thereby pushing the limits of its overall performance. We first demonstrated that using Pass@k as the reward can effectively enhance the model's exploration ability. Next, to improve training efficiency and effectiveness, we introduce the bootstrap sampling and analytical derivation to optimize the Pass@k Training procedure. After that, to better understand the inner mechanism of Pass@k Training, we proposed five research questions from different aspects to answer why the Pass@k Training works and what benefits it brings. Moreover, to make Pass@k Training employable in real-world applications, we conducted an exploration into designing customized advantage functions, *i.e., implicit reward design*, showing remarkable effectiveness. We consider it a promising direction for future research.

## ETHICS STATEMENT

During the research process, we strictly adhered to academic standards and ethical guidelines. All the data used in our experiments strictly follows the ethical standards, *i.e.,* it contained no personal privacy information, no content that violates human values, and no biased or offensive material. Our research aims to enhance the intelligence of large language models, with the goal of enabling AI technologies to better assist all humankind, contributing to society and human well-being. To ensure the validity of our research, we conducted rigorous and comprehensive experiments that are provided in the Appendix, holding our work to the highest standards. Besides, we only use the large language models to examine and correct the grammar mistakes of our paper, and we manually review the content generated by the AI assistants to ensure the rigor and accuracy of our paper.

## REPRODUCIBILITY STATEMENT

To ensure the reproducibility of our work, we provide a detailed description of the algorithmic details in Sec. 2, along with pseudocode for its implementation in Appendix D. Moreover, we present all the details about the implementation of our experiments in Appendix B, including the hyper-parameters of training and evaluation and the details of the evaluation benchmarks. Furthermore, we also provide the code of our approach and the data used in our experiments in the Supplementary Material. We believe the above information can help readers and researchers to reproduce our work.

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

## A  RELATED WORK

**Reinforcement Learning with Verifiable Rewards.** To unleash the potential of LLM reasoning ability, DeepSeek directly employs reinforcement learning with verifiable rewards (RLVR) on DeepSeek-V3, obtaining the large reasoning model DeepSeek-R1-Zero (DeepSeek-AI et al., 2025), which can perform the reasoning process with complex reasoning actions (*e.g.,* reflection and verification). Given the success of the DeepSeek-R1, a surge of studies (Chen et al., 2025b; Zeng et al., 2025; Hu et al., 2025a) have explored the effectiveness of RLVR on the popular open-source LLMs, like Qwen (Yang et al., 2024), Mistral (Jiang et al., 2023), and LLaMA (Dubey et al., 2024). Moreover, the RLVR training paradigm can help LLMs to control their reasoning time (Aggarwal & Welleck, 2025), switch the reasoning pattern (Chung et al., 2025; Wu et al., 2025), enhance the specific performance metric (Tang et al., 2025), and enhance their abilities without supervision (Zuo et al., 2025; Hu et al., 2025b). However, recent work points out that the popular RLVR algorithms (*e.g.,* PPO (Schulman et al., 2017) and GRPO (Shao et al., 2024)) still face serious challenges, like training instability, model collapse, and reward noise (Walder & Karkhanis, 2025; Liu et al., 2025c; Yu et al., 2025; Liu et al., 2025a; Zhu et al., 2025). To mitigate these issues, existing researches propose the optimization on the rollout strategy (Yu et al., 2025), objective function design (Walder & Karkhanis, 2025; Liu et al., 2025c;a), and data selection (Zhu et al., 2025). Specifically, previous work (Walder & Karkhanis, 2025) utilizes Pass@k as the reward on the policy gradient (Williams, 1992) to encourage models to solve hard problems. However, the intrinsic connection between Pass@k RLVR training and LLM exploration ability has not been fully recognized. Thus, we further adopt the Pass@k metric in GRPO and its variants through three approaches (Fig. 9), and derive the analytical solution of advantage values of Pass@k reward in RLVR training. Moreover, according to empirical experiments and theoretical analysis, we discuss the benefits of Pass@k Training in balancing the exploration and exploitation abilities of LLMs during the RLVR training procedure, showing the huge potential of Pass@k RLVR training and pointing out the promising future research directions.

**Effective Exploration in Test-time Scaling.** Recently, test-time scaling has been proposed to improve the performance of LLMs by consuming more computational resources at inference time (Zhang et al., 2025). Since the LLMs continuously leverages exploration-derived experience to optimize its performance, effective exploration is important and necessary during the test-time scaling process (Hou et al., 2025; Luo et al., 2025). However, existing work reveals that the exploration ability is limited by the corresponding base model, hindering the continuous scaling of model performance (Yue et al., 2025). To mitigate this issue, previous work proposed several approaches, including achieved by adjusting the sampling hyper-parameters (Chen et al., 2025b; Hu et al., 2025a; Hou et al., 2025), performing self-verification and self-reflection (Jiang et al., 2025; Liu et al., 2025b; Sareen et al., 2025), or leveraging external models to verify the reasoning process (Liu et al., 2025d; Zha et al., 2025). Beyond these approaches from an external perspective of the model, it is equally important to explore the model's exploration capability through its internal mechanisms. Current studies start from the perspective of the entropy of policy distribution, pointing out that entropy can indicate the exploration ability of LLMs (Cheng et al., 2025; Cui et al., 2025) and high-entropy tokens are vital for model optimization (Wang et al., 2025). Based on these findings, training the critical tokens (Wang et al., 2025) and adding regularization (Hou et al., 2025; Liu et al., 2025a) are employed in the RLVR training process to avoid the degradation of the exploration capability of LLMs. Further, several studies focus on enhancing the exploration abilities of LLMs by selecting useful sampled experience (Zhu et al., 2025; Setlur et al., 2025), integrating entropy into advantage estimation (Cui et al., 2025).

## B  EXPERIMENT SETUP

### B.1  DETAILS OF DOWNSTREAM TASKS

In this section, we present detailed information of each downstream evaluation task.

**Maze.** We follow the framework proposed by previous work to synthesize the different sizes of mazes. Each maze is represented by text, containing $n$ rows and $n$ columns, a total of $n*n$ characters.

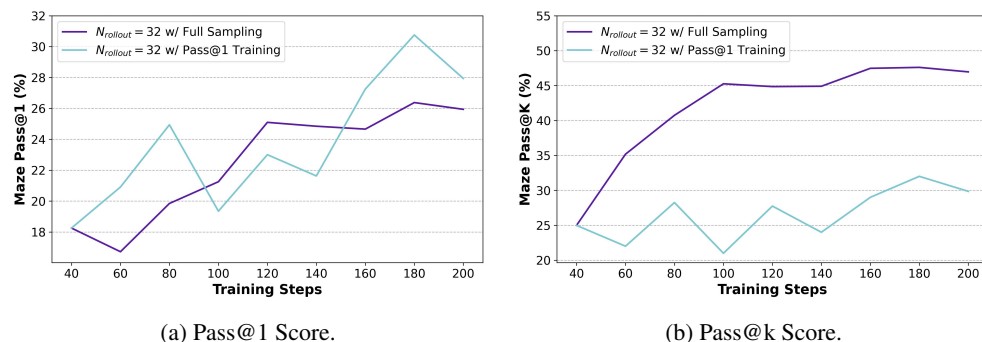

(a) Pass@1 Score.        (b) Pass@k Score.

Figure 7: Training progress on Maze task of Pass@1 Training and Pass@k Training with Full Sampling.

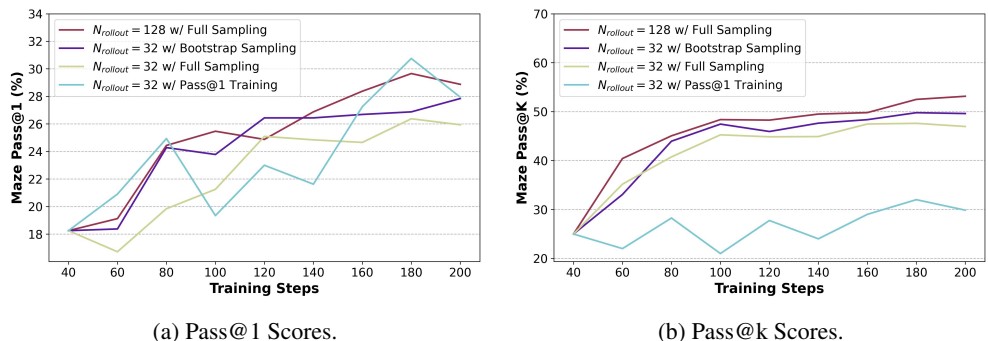

(a) Pass@1 Scores.        (b) Pass@k Scores.

Figure 8: Training progress on Maze task of Pass@1 Training and Pass@k Training with Bootstrap Sampling under various $N_{\text{rollout}}$.

Concretely, each of them is one of the following four characters "S", "E", "*", and ".", denoting the start point, destination, available place, and unavailable place, respectively. Given the maze, LLMs can first generate the thought or reasoning process and then generate the final answer, which includes one of the four actions "U", "D", "L", and "R", refering to moving up, down, left, and right, respectively. For training data, we construct the mazes with sizes of $9 \times 9$, $11 \times 11$, $13 \times 13$, and $15 \times 15$, to increase the diversity of training data. For test data, to evaluate the generalization of the RLVR process, we not only conduct the same sizes of the mazes with the training dataset, but also collect the mazes with sizes of $7 \times 7$, $17 \times 17$, $19 \times 19$, and $21 \times 21$. To ensure the validity of the experiment, we performed strict deduplication operations after generating the training and test data. The statistical information of the datasets is presented in Table 7. For better understanding, we present a test instance in Figure 10. To present the empirical insights more clearly, we only showed the results of the $9 \times 9$ maze in the above text, and the remaining results are presented in Appendix F.4.

Table 7: The statistical information of the Maze task.

|  | $7 \times 7$ | $9 \times 9$ | $11 \times 11$ | $13 \times 13$ | $15 \times 15$ | $17 \times 17$ | $19 \times 19$ | $21 \times 21$ |
|---|---|---|---|---|---|---|---|---|
| Training Set | - | 10,000 | 10,000 | 10,000 | 10,000 | - | - | - |
| Test Set | 75 | 100 | 100 | 100 | 100 | 100 | 100 | 100 |

**Enigmata.** To assess the reasoning and logical abilities of LLMs, Enigmata proposed a comprehensive benchmark that includes the 36 categories of synthetic verifiable puzzles of 7 primary categories, including Crypto Puzzle, Arithmetic Puzzle, Logic Puzzle, Grid Puzzle, Graph Puzzle, Search Puzzle, and Sequential Puzzle. Each category can assess different abilities of LLMs. For better understanding, we present a test instance in Figure 11.

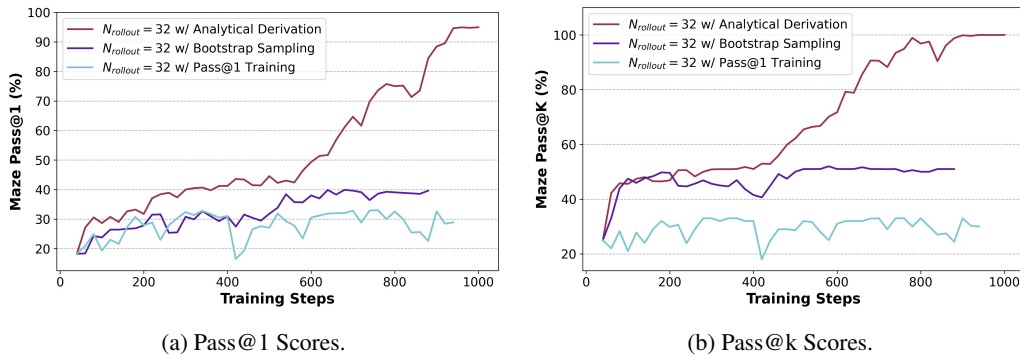

(a) Pass@1 Scores.                    (b) Pass@k Scores.

Figure 9: Training progress on Maze task of Pass@1 Training and Pass@k Training with Analytical Derivation and Bootstrap Sampling.

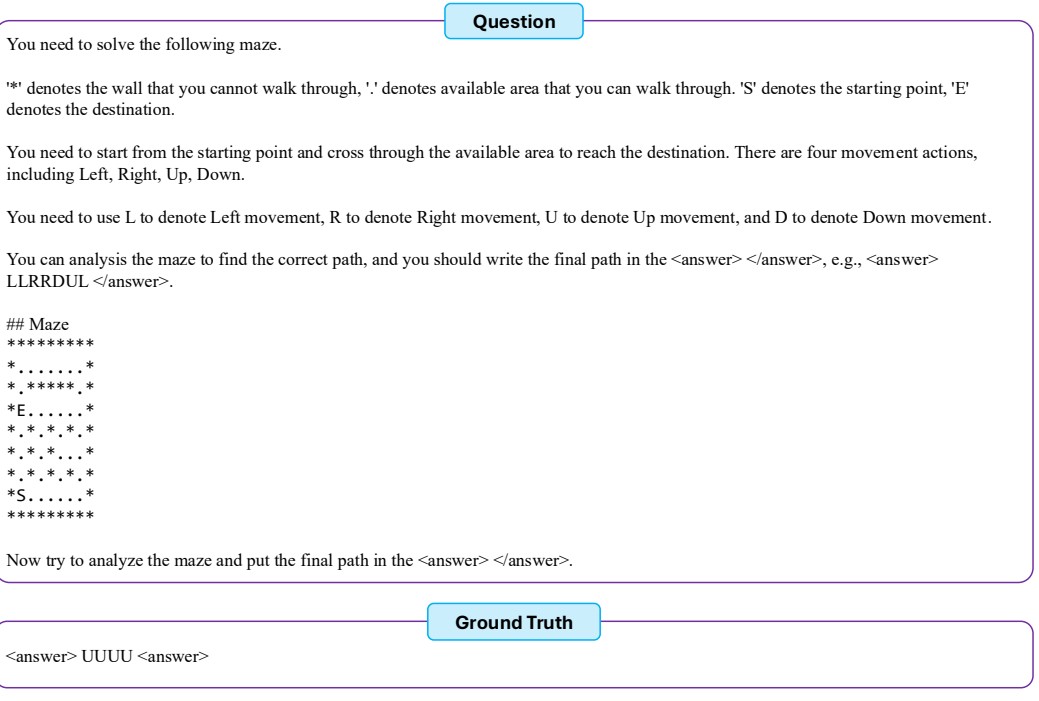

Figure 10: An example of **Maze** Task.

**MathVision.** MathVision selects 3,040 high-quality problems from human math competitions, each accompanied by relevant images. Solving these problems requires both careful interpretation of the visual information and rigorous mathematical reasoning. MathVision provides a benchmark for assessing a model's multimodal understanding as well as its ability to perform rigorous mathematical reasoning. For better understanding, we present a test instance in Figure 12.

**MMMU.** MMMU includes college-level reasoning and comprehension tasks across six academic subjects, including Art & Design, Business, Science, Health & Medicine, Humanities & Social Science, and Tech & Engineering. Moreover, MMMU includes a wide range of image types, enabling a comprehensive assessment of a model's capability to process and reason over different forms of visual information. For better understanding, we present a test instance in Figure 13.

**Question**

Apply a function to the final input list to generate the output list. Use any preceding inputs and outputs as examples to find what is the function used. All example outputs have been generated using the same function.

### Response Format:
-   Please output your answer within a code block (```), formatted as a list of numbers, for example: ``` [0, 2, 3] ```

# Examples

Example 1:
[6, 4] -> [6, 8, 4]

Example 2:
[8, 3, 2, 0, 9, 7] -> [8, 5, 3, 2, 0, 9, 7]

Example 3:
[1, 2, 6, 0, 9, 3] -> [1, 5, 2, 6, 0, 9, 3]

Example 4: [9, 7, 8] -> [9, 8, 7, 8]

Example 5:
[1, 9, 6, 5, 0, 3, 8, 4, 7, 2] -> [1, 5, 9, 6, 5, 0, 3, 8, 4, 7, 2]

Example 6:
[9, 8] -> [9, 8, 8]

 # Test Problem:
[7, 4, 6, 8, 0, 1, 3] ->

**Ground Truth**

```[7, 5, 4, 6, 8, 0, 1, 3]```

Figure 11: An example of **Enigmata** Task.

**Question**

Which number should be written in place of the question mark?

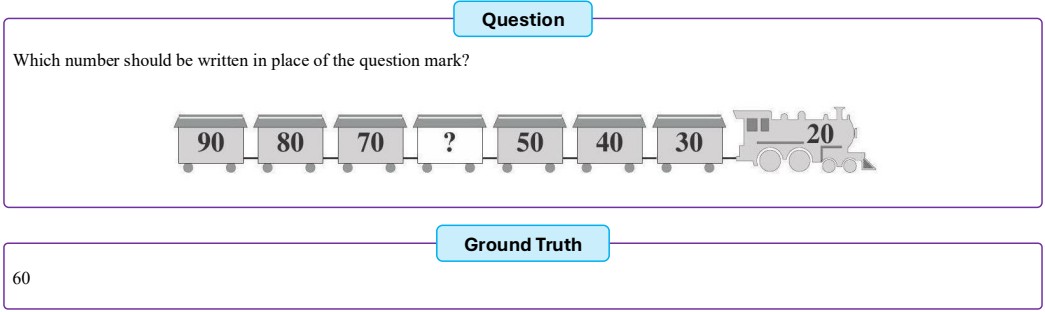

**Ground Truth**

60

Figure 12: An example of **MathVision** Task.

## B.2 IMPLEMENTATION DETAILS

**Training.** In our experiment, we adapt Qwen2.5-7B-Instruct and Qwen2.5-32B-Instruct as the backbone model and train it through DAPO. To enhance the efficiency of the training process, we only retain the clip-higher (i.e., $\varepsilon_{\text{low}} = 0.2$ and $\varepsilon_{\text{high}} = 0.28$) and token-level policy gradient loss, and remove other optimizations. For the training hyper-parameters, we set the learning rate for the policy model as $1 \times 10^{-6}$ with 10 warmup steps, and employ 128, 32, and 32 as prompt batch size $BS_{\text{prompt}}$, mini-batch size $BS_{\text{mini}}$, and rollout times $n_{\text{rollout}}$, respectively. For the reward, the responses that pass the verification (named as positive responses) will be assigned the positive reward $R_{\text{pos}} = 1$, while the other responses (named as negative responses) will be endowed with the negative reward $R_{\text{neg}} = 0$. Additionally, we do not employ any regularization methods, such as KL or Entropy regularization. Besides, for the maximum length of the model response, we set it as 10240 for the puzzle task and 6144 for the mathematical and maze tasks. For other hyper-parameters that are not mentioned above, we adopt the default settings in the verl framework (Sheng et al., 2025).

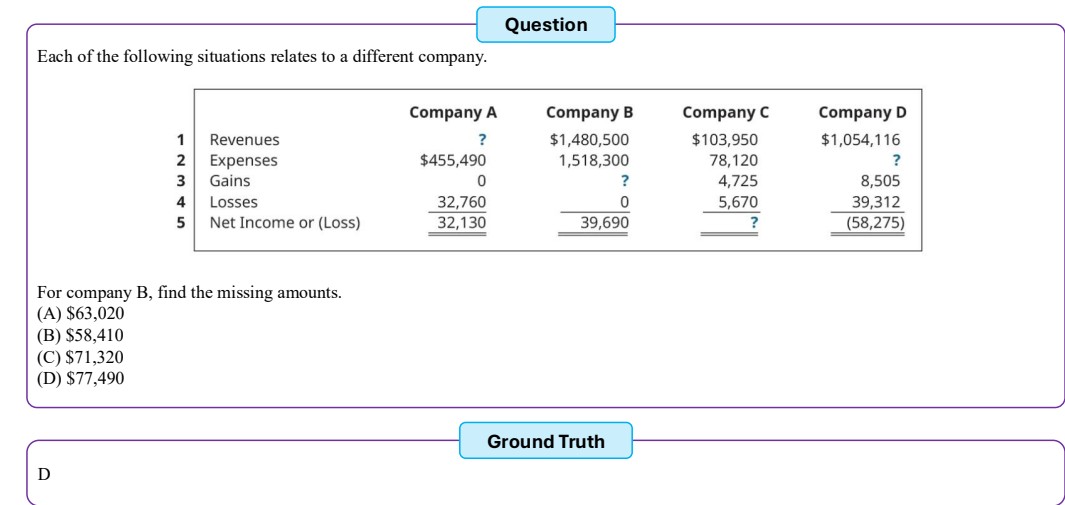

Figure 13: An example of **MMMU** Tasks.

For the training data, we collect the questions that share the same domain as the downstream tasks, *i.e.,* puzzle, maze, and mathematics.

**Evaluation.** To evaluate the performance of LLMs, we adopt 1.0 and 0.95 as the Temperature and Top_P. For each question, we sample 32 responses from LLMs for the Maze task and sample 8 responses from LLMs for other tasks, and then utilize the sampled response to compute the Pass@1 and Pass@k scores.

## C   DETAILS OF ANALYTICAL DERIVATION

We present the details of the analytical derivation procedure mentioned in Sec. 2.4, including the derivation of the average of the group reward, standard deviation of the group reward, and response-relative advantage.

### C.1   DERIVATION OF $\hat{A}_{\text{POS}} \geq 0$

Since $\bar{R}^{\text{group}}$ denotes the average value of the reward of each group, it will be less than or equal to the maximum reward of the group, *i.e.,*

$$\bar{R}^{\text{group}} \leq \max(R_{\text{pos}}, R_{\text{neg}}) = R_{\text{pos}} = 1. \tag{10}$$

Based on Eq. 10, we can obtain that $1 - \bar{R}^{\text{group}} \geq 0$. Given that $\sigma^{\text{group}} > 0$, we can derive that,

$$\hat{A}_{\text{pos}} = \left(1 - \bar{R}^{\text{group}}\right) \times (\sigma^{\text{group}})^{-1} \geq 0. \tag{11}$$

### C.2   DERIVATION OF $\hat{A}_{\text{NEG}} \leq 0$

Based on Eq. 6 and Eq. 12, we can derive that,

$$\hat{A}_{\text{neg}} = \left(1 - \bar{R}^{\text{group}} - \frac{\binom{N_{\text{neg}} - 1}{k-1}}{\binom{N_{\text{rollout}} - 1}{k-1}}\right) \times (\sigma^{\text{group}})^{-1} = \left(\frac{\binom{N_{\text{neg}}}{k}}{\binom{N_{\text{rollout}}}{k}} - \frac{\binom{N_{\text{neg}} - 1}{k-1}}{\binom{N_{\text{rollout}} - 1}{k-1}}\right) \times (\sigma^{\text{group}})^{-1}. \tag{12}$$

In the following, we derive whether $\frac{\binom{N_{\text{neg}}}{k}}{\binom{N_{\text{rollout}}}{k}} - \frac{\binom{N_{\text{neg}} - 1}{k-1}}{\binom{N_{\text{rollout}} - 1}{k-1}} \leq 0$, *i.e.,*

$$\frac{\binom{N_{\text{neg}}}{k}}{\binom{N_{\text{rollout}}}{k}} - \frac{\binom{N_{\text{neg}} - 1}{k-1}}{\binom{N_{\text{rollout}} - 1}{k-1}} = \binom{N_{\text{neg}}}{k} \times \binom{N_{\text{rollout}} - 1}{k - 1} - \binom{N_{\text{neg}} - 1}{k - 1} \times \binom{N_{\text{rollout}} - 1}{k - 1} \tag{13}$$

Since $\binom{N_{\text{rollout}}}{k} \times \binom{N_{\text{rollout}}-1}{k-1} > 0$, we only need to derive $\binom{N_{\text{neg}}}{k} \times \binom{N_{\text{rollout}}-1}{k-1} - \binom{N_{\text{neg}}-1}{k-1} \times \binom{N_{\text{rollout}}-1}{k-1} \leq 0$.

$$
\begin{aligned}
& \binom{N_{\text{neg}}}{k} \times \binom{N_{\text{rollout}}-1}{k-1} - \binom{N_{\text{neg}}-1}{k-1} \times \binom{N_{\text{rollout}}-1}{k-1} \\
= & \frac{N_{\text{neg}}! \times (N_{\text{rollout}}-1)!}{k!(N_{\text{neg}}-k)! \times (k-1)!(N_{\text{rollout}}-k)!} - \frac{(N_{\text{neg}}-1)! \times N_{\text{rollout}}!}{(k-1)!(N_{\text{neg}}-k)! \times k!(N_{\text{rollout}}-k)!} \\
= & \frac{(N_{\text{neg}}-1)!(N_{\text{rollout}}-1)!}{k!(k-1)!(N_{\text{neg}}-k)!(N_{\text{rollout}}-k)!} \times (N_{\text{neg}} - N_{\text{rollout}}).
\end{aligned}
\tag{14}
$$

Since

$$
\frac{(N_{\text{neg}}-1)!(N_{\text{rollout}}-1)!}{k!(k-1)!(N_{\text{neg}}-k)!(N_{\text{rollout}}-k)!} > 0,
\tag{15}
$$

and

$$
N_{\text{neg}} \leq N_{\text{rollout}},
\tag{16}
$$

we have $\binom{N_{\text{neg}}}{k} \times \binom{N_{\text{rollout}}-1}{k-1} - \binom{N_{\text{neg}}-1}{k-1} \times \binom{N_{\text{rollout}}-1}{k-1} \leq 0$. Therefore, we can derive that

$$
\hat{A}_{\text{neg}} \leq 0.
\tag{17}
$$

## C.3 DERIVATION OF THE AVERAGE OF GROUP REWARD

$$
\begin{aligned}
\bar{R}^{\text{group}} &= \frac{1}{N_{\text{total}}^{\text{group}}} \times \left( N_{\text{pos}}^{\text{group}} \times R_{\text{pos}} + N_{\text{neg}}^{\text{group}} \times R_{\text{neg}} \right) \\
&= \frac{1}{\binom{N_{\text{rollout}}}{K}} \times \left( \left( \binom{N_{\text{rollout}}}{K} - \binom{N_{\text{neg}}}{K} \right) \times 1 + \left( \binom{N_{\text{neg}}}{K} \right) \times 0 \right) \\
&= 1 - \frac{\binom{N_{\text{neg}}}{K}}{\binom{N_{\text{rollout}}}{K}}.
\end{aligned}
\tag{18}
$$

## C.4 DERIVATION OF THE STANDARD DEVIATION OF GROUP REWARD

$$
\begin{aligned}
\sigma^{\text{group}} &= \sqrt{\frac{1}{N_{\text{rollout}}} \times \left( N_{\text{pos}}^{\text{group}} \times \left( \bar{R}^{\text{group}} - R_{\text{pos}} \right)^2 + N_{\text{neg}}^{\text{group}} \times \left( \bar{R}^{\text{group}} - R_{\text{neg}} \right)^2 \right)} \\
&= \sqrt{\frac{\left( \binom{N_{\text{rollout}}}{K} - \binom{N_{\text{neg}}}{K} \right) \times \left( 1 - \frac{\binom{N_{\text{neg}}}{K}}{\binom{N_{\text{bootstrap}}}{K}} - 1 \right)^2 + \binom{N_{\text{neg}}}{K} \times \left( 1 - \frac{\binom{N_{\text{neg}}}{K}}{\binom{N_{\text{rollout}}}{K}} - 0 \right)^2}{\binom{N_{\text{rollout}}}{K}}} \\
&= \sqrt{\left( 1 - \frac{\binom{N_{\text{neg}}}{K}}{\binom{N_{\text{bootstrap}}}{K}} \right) \times \left( \frac{\binom{N_{\text{neg}}}{K}}{\binom{N_{\text{rollout}}}{K}} \right)^2 + \frac{\binom{N_{\text{neg}}}{K}}{\binom{N_{\text{rollout}}}{K}} \times \left( 1 - \frac{\binom{N_{\text{neg}}}{K}}{\binom{N_{\text{rollout}}}{K}} \right)^2} \\
&= \sqrt{\left( 1 - \frac{\binom{N_{\text{neg}}}{K}}{\binom{N_{\text{rollout}}}{K}} \right) \times \left( \frac{\binom{N_{\text{neg}}}{K}}{\binom{N_{\text{rollout}}}{K}} \right) \times \left( 1 - \frac{\binom{N_{\text{neg}}}{K}}{\binom{N_{\text{rollout}}}{K}} + \frac{\binom{N_{\text{neg}}}{K}}{\binom{N_{\text{rollout}}}{K}} \right)} \\
&= \sqrt{\left( 1 - \frac{\binom{N_{\text{neg}}}{K}}{\binom{N_{\text{rollout}}}{K}} \right) \times \left( \frac{\binom{N_{\text{neg}}}{K}}{\binom{N_{\text{rollout}}}{K}} \right)} \\
&= \sqrt{\bar{R}^{\text{group}} \times \left( 1 - \bar{R}^{\text{group}} \right)}.
\end{aligned}
\tag{19}
$$

## C.5 DERIVATION OF THE RESPONSE-RELATIVE ADVANTAGE

$$
\begin{aligned}
\hat{A}_{\mathrm{pos}} &= \frac{1}{\binom{N_{\mathrm{rollout}}-1}{K-1}} \times \left( \binom{N_{\mathrm{rollout}}-1}{K-1} \times \hat{A}_{pos}^{\mathrm{group}} + 0 \times \hat{A}_{neg}^{\mathrm{group}} \right) \\
&= \frac{1 - \bar{R}^{\mathrm{group}}}{\sigma^{\mathrm{group}}}. \\
\hat{A}_{\mathrm{neg}} &= \frac{1}{\binom{N_{\mathrm{rollout}}-1}{K-1}} \times \left( \left( \binom{N_{\mathrm{rollout}}-1}{K-1} - \binom{N_{\mathrm{neg}}-1}{K-1} \right) \times \hat{A}_{pos}^{\mathrm{group}} + \binom{N_{\mathrm{neg}}-1}{K-1} \times \hat{A}_{neg}^{\mathrm{group}} \right) \\
&= \left( \left( 1 - \frac{\binom{N_{\mathrm{neg}}-1}{K-1}}{\binom{N_{\mathrm{rollout}}-1}{K-1}} \right) \times \hat{A}_{\mathrm{pos}}^{\mathrm{group}} + \frac{\binom{N_{\mathrm{neg}}-1}{K-1}}{\binom{N_{\mathrm{rollout}}-1}{K-1}} \times \hat{A}_{\mathrm{neg}}^{\mathrm{group}} \right) \\
&= \left( \left( 1 - \frac{\binom{N_{\mathrm{neg}}-1}{K-1}}{\binom{N_{\mathrm{rollout}}-1}{K-1}} \right) \times \frac{1 - \bar{R}^{\mathrm{group}}}{\sigma^{\mathrm{group}}} + \frac{\binom{N_{\mathrm{neg}}-1}{K-1}}{\binom{N_{\mathrm{rollout}}-1}{K-1}} \times \left( -\frac{\bar{R}^{\mathrm{group}}}{\sigma^{\mathrm{group}}} \right) \right) \\
&= \left( 1 - \bar{R}^{\mathrm{group}} - \frac{\binom{N_{\mathrm{neg}}-1}{K-1}}{\binom{N_{\mathrm{rollout}}-1}{K-1}} \right) \times (\sigma^{\mathrm{group}})^{-1}.
\end{aligned}
\tag{20}
$$

## D PSEUDO CODE FOR PASS@K TRAINING

We present the pseudo code for Pass@k Training with full sampling (Algorithm 1), bootstrap sampling (Algorithm 2), and analytical derivation (Algorithm 3).

---

**Algorithm 1:** The Pseudo Code for Pass@k Training with Full Sampling.

---

**Input** : A tensor of reward $\mathcal{R} \in \mathbb{R}^{N_{\mathrm{rollout}}}$ of the responses for the problem, the number of the rollouted responses $N_{\mathrm{rollout}}$, and the $k$ for Pass@k metric.

**Output:** A tensor of estimated advantages of the responses for this problem $\hat{\mathcal{A}} \in \mathbb{R}^{N_{\mathrm{rollout}}}$.

1 # Construct the groups and discard the redundant instances.
2 Separate $\mathcal{R} \in \mathbb{R}^{N_{\mathrm{rollout}}}$ into $\lfloor \frac{N_{\mathrm{rollout}}}{K} \rfloor$ group and each group contains $k$ instances.
3 Compute the reward of the groups $\mathcal{R}^{\mathrm{group}} \in \mathbb{R}^{\lfloor \frac{N_{\mathrm{rollout}}}{K} \rfloor}$ using Eq. 3.

4 # Follow GRPO advantage estimation method to compute group-relative advantage.
5 Compute the average reward of the groups $\bar{R}^{\mathrm{group}}$ using the left part of Eq. 1.
6 Compute the standard deviation of the groups $\sigma^{\mathrm{group}}$ using the middle part of Eq. 1.
7 Based on $\bar{R}^{\mathrm{group}}$ and $\sigma^{\mathrm{group}}$, compute the group advantage $\hat{\mathcal{A}}^{\mathrm{group}}$ using the right part of Eq. 1.

8 # Compute response advantage.
9 Assign the $\hat{\mathcal{A}}^{\mathrm{group}}$ to the responses that the group contains, obtaining the response advantage $\hat{\mathcal{A}}$.

---

## E CURVES OF ADVANTAGE FUNCTION

We present the curves of the advantage function of different training approaches in Figure 14, including Pass@k Training w/o easy problems, Pass@k Training w/ combination, exceeding Pass@k Training, and combination training.

## F EXPERIMENTS ON VARIOUS LLMS AND TASKS

In this section, to further verify the effectiveness of Pass@k Training, we first present the significance test for the results in Sec. 3.5, and provide the performance of various LLMs trained through Pass@k Training on Mathematical Tasks (*i.e.,* AIME 2024 (AIME2024, 2024), AIME 2025 (AIME2025, 2025), and OlymMATH (Sun et al., 2025)) and Synthetic Puzzle Task (*i.e.,* Enigmata (Chen et al., 2025a)).

---

**Algorithm 2:** The Pseudo Code for Pass@k Training with Bootstrap Sampling.

---

**Input** : A tensor of reward $\mathcal{R} \in \mathbb{R}^{N_{\text{rollout}}}$ of the responses for the problem, the number of the rollouted responses $N_{\text{rollout}}$, and the $k$ for Pass@k metric.

**Output:** A tensor of estimated advantages of the responses for this problem $\hat{\mathcal{A}} \in \mathbb{R}^{N_{\text{rollout}}}$.

---

1 # Construct the groups through bootstrap sampling.
2 **for** $i$ *from* $1$ *to* $N^{group}$ **do**
3     Randomly sample $k$ instances from $\mathcal{R}$ to construct the $i$-th group.
4     Compute the reward of $i$-th group using Eq. 3.
5 Obtain the reward of the groups $\mathcal{R}^{\text{group}} \in \mathbb{R}^{N^{\text{group}}}$.

6 # Follow GRPO advantage estimation method to compute group advantage.
7 Compute the average reward of the groups $\bar{R}^{\text{group}}$ using the left part of Eq. 1.
8 Compute the standard deviation of the groups $\sigma^{\text{group}}$ using the middle part of Eq. 1.
9 Based on $\bar{R}^{\text{group}}$ and $\sigma^{\text{group}}$, compute the group advantage $\hat{\mathcal{A}}^{\text{group}}$ using the right part of Eq. 1.

10 # Calculate response advantage.
11 Based on $\hat{\mathcal{A}}^{\text{group}}$, compute response advantage $\hat{\mathcal{A}}$ using Eq. 4.

---

---

**Algorithm 3:** The Pseudo Code for Pass@k Training with Analytical Derivation.

---

**Input** : A tensor of reward $\mathcal{R} \in \mathbb{R}^{N_{\text{rollout}}}$ of the responses for the problem, the number of the rollouted responses $N_{\text{rollout}}$, and the $k$ for Pass@k metric.

**Output:** A tensor of estimated advantages of the responses for this problem $\hat{\mathcal{A}} \in \mathbb{R}^{N_{\text{rollout}}}$.

---

1 # Calculate the average and standard deviation of the group reward scores.
2 Compute the average reward of the groups $\bar{R}^{\text{group}}$ using the left part of Eq. 6.
3 Compute the standard deviation of the groups $\sigma^{\text{group}}$ using the right part of Eq. 6.

4 # Calculate response advantage.
5 Compute the advantage of the positive responses $\hat{A}_{\text{pos}}$ using Eq. 8.
6 Compute the advantage of the negative responses $\hat{A}_{\text{neg}}$ using Eq. 12.
7 Based on $\hat{A}_{\text{pos}}$, $\hat{A}_{\text{neg}}$, and $\mathcal{R}$, assign the advantage to each instance, obtaining response advantage $\hat{\mathcal{A}}$.

---

### F.1 SIGNIFICANCE TEST

To strengthen the paper's credibility, we provided the mean and variance of multiple runs for different algorithms on the Enigmata task in Table 8.

By comparing the performance of "P@1 T." and "P@k T." on the Pass@k score, we find that the difference in their means is more than three times the variance, indicating with 99.7% confidence that Pass@k training achieves higher Pass@k scores than Pass@1 training. Similarly, by comparing "+P@1 T." and "+P@k T. + P@1 T." on the Pass@1 score, we find that the difference in their means exceeds twice the variance, providing 95.4% confidence that our method achieves better Pass@1 scores than the traditional Pass@1 training approach.

Similarly, by comparing the performance of "GPT-4o-1120" and "+P@k T. + P@1 T.", we find that the difference in their average Pass@1 score also exceeds twice the variance, demonstrating that our method enables the 7B model to outperform larger models.

### F.2 PASS@K TRAINING ON MATHEMATICAL TASKS

We follow the experiment settings described in Appendix B.2 to perform Pass@k Training on LLaMA models (Dubey et al., 2024) (*i.e.,* LLaMA3.2-3B-Instruct and LLaMA3.1-8B-Instruct) and DeepSeek-R1-Distill-Qwen (DeepSeek-AI et al., 2025) (*i.e.,* 1.5B and 7B version). For LLaMA models, we set the maximum prompt length and response length as 2048 and 6144, respectively. For DeepSeek-R1-Distill-Qwen, we extend the response length to 10240. Specifically, to adapt the LLMs to the mathematical tasks, we adopt the training data used in previous work (Chen et al.,

Table 8: The mean and variance of multiple runs for different algorithms on the Enigmata task.

| Methods | Pass@1 Score | | Pass@k Score | |
|---|---|---|---|---|
| | mean@8 | std@8 | mean@8 | std@8 |
| GPT-4o-1120 | 14.2 | - | - | - |
| Qwen2.5-7B-Instruct + P@1 T. | 12.9 | 1.91 | 21.3 | 0.36 |
| Qwen2.5-7B-Instruct + P@k T. | 17.9 | 4.70 | 29.8 | 0.73 |
| Qwen2.5-7B-Instruct + P@k T. + P@1 T. | 30.8 | 7.44 | 40.6 | 0.28 |

2025b) during the RLVR training process. Besides, we follow the settings in Appendix B.2 to perform the evaluation, and present the results in Table 9. Since the single turn of Pass@k Training followed by Pass@1 Training can significantly improve the Pass@1 performance of LLMs, we conduct the experiment about multiple turns of the above training process in Table 9, named as "(P@k T. + P@1 T.) × 2".

Table 9: Pass@1/Pass@k Performance on mathematical tasks of LLaMA and DeepSeek-R1-Distill-Qwen models trained through different RLVR approaches. "P@1 T." and "P@k T." denote the Pass@1 Training and Pass@k Training with analytical derivation, respectively. "(P@k T. + P@1 T.) × 2" refers to that the process of Pass@k Training followed by Pass@1 Training is repeated twice.

| | AIME 2024 | AIME 2025 | OlymMATH-Easy | OlymMATH-Hard | Avg. |
|---|---|---|---|---|---|
| **RLVR on LLaMA3.2-3B-Instruct (Pass@1/Pass@k)** | | | | | |
| Baseline | 1.5/17.3 | 0.1/2.1 | 1.7/14.4 | 1.1/9.2 | 1.1/10.8 |
| + P@1 T. | 13.6/26.7 | 1.1/6.6 | 3.8/4.0 | 2.0/6.3 | 5.1/10.9 |
| + P@k T. | 12.7/32.0 | 1.7/12.9 | 3.7/8.8 | 1.7/7.7 | 5.0/**15.4** |
| + P@k T. + P@1 T. | 14.6/32.1 | 1.3/8.6 | 4.1/7.7 | 2.0/7.5 | **5.5**/14.0 |
| **RLVR on LLaMA3.1-8B-Instruct (Pass@1/Pass@k)** | | | | | |
| Baseline | 3.4/17.9 | 0.2/4.3 | 0.8/7.5 | 0.5/7.6 | 1.0/9.3 |
| + P@1 T. | 4.4/32.1 | 0.9/7.7 | 1.4/4.1 | 1.1/6.2 | 2.0/12.5 |
| + P@k T. | 7.1/40.0 | 1.8/10.6 | 1.5/8.9 | 1.4/8.2 | 3.0/**17.0** |
| + P@k T. + P@1 T. | 8.7/29.7 | 0.9/8.7 | 1.8/7.9 | 1.6/6.8 | **3.3**/13.3 |
| **RLVR on DeepSeek-R1-Distill-Qwen-1.5B (Pass@1/Pass@k)** | | | | | |
| Baseline | 22.7/61.4 | 20.5/37.2 | 6.6/36.7 | 0.6/5.2 | 12.6/35.1 |
| + P@1 T. | 36.7/76.0 | 28.8/49.4 | 16.7/51.7 | 2.5/17.5 | 21.2/48.7 |
| + P@k T. | 36.5/79.3 | 27.0/55.5 | 17.6/59.3 | 2.4/17.4 | 20.9/52.9 |
| + P@k T. + P@1 T. | 42.3/71.7 | 30.4/57.8 | 20.7/60.5 | 3.4/18.9 | 24.2/52.2 |
| + (P@k T. + P@1 T.) × 2 | 44.2/77.2 | 31.5/57.7 | 22.6/62.7 | 4.4/21.2 | **25.7/54.7** |
| **RLVR on DeepSeek-R1-Distill-Qwen-7B (Pass@1/Pass@k)** | | | | | |
| Baseline | 43.2/80.9 | 31.5/59.1 | 22.2/66.0 | 1.4/13.7 | 24.6/54.9 |
| + P@1 T. | 48.5/79.5 | 35.5/59.4 | 27.9/69.1 | 3.1/22.5 | 28.8/57.6 |
| + P@k T. | 48.2/80.9 | 36.5/66.7 | 28.1/72.7 | 3.3/23.3 | 29.0/**60.9** |
| + P@k T. + P@1 T. | 50.3/81.0 | 39.3/61.9 | 32.3/68.9 | 3.5/22.7 | **31.4**/58.6 |

### F.3 PASS@K TRAINING ON ENIGMATA TASK

We follow the experiment settings described in Appendix B.2 to perform Pass@k Training on various LLMs (*i.e.,* LLaMA3.2-3B-Instruct (Dubey et al., 2024) and LLaMA3.1-8B-Instruct (Dubey et al., 2024)), and set the maximum of the prompt length and response length as 4096 and 4096, respectively. The results are presented in Table 10. For evaluation, we follow the settings described in Appendix B.2.

### F.4 PASS@K TRAINING ON MAZE TASK

In this part, we present the full results of Pass@k Training on the Maze task in Table 11. Without any RLVR training, it is really difficult for the model to solve the Maze task. Thus, we do not report the performance of the backbone model.

Table 10: Enigmata Pass@1/Pass@k Performance of LLaMA models trained on different RLVR approaches. "P@1 T." and "P@k T." denote the Pass@1 Training and Pass@k Training with analytical derivation, respectively.

| | Crypto | Arithmetic | Logic | Grid | Graph | Search | Sequential | Overall |
|---|---|---|---|---|---|---|---|---|
| **RLVR on LLaMA3.2-3B-Instruct (Pass@1/Pass@k)** | | | | | | | | |
| Baseline | 0.0/0.0 | 0.2/1.6 | 19.3/44.3 | 2.7/4.7 | 1.7/8.8 | 5.4/11.1 | 0.4/1.8 | 3.1/7.3 |
| + P@1 T. | 0.0/0.0 | 0.2/1.3 | 19.7/27.0 | 17.4/18.0 | 5.3/12.8 | 12.9/14.5 | 9.8/10.7 | 11.1/13.0 |
| + P@k T. | 0.0/0.0 | 0.2/0.7 | 22.0/31.0 | 17.3/18.1 | 6.1/14.2 | 12.2/16.6 | 10.7/12.0 | 11.5/**14.1** |
| + P@k T. + P@1 T. | 0.0/0.0 | 0.4/2.2 | 22.8/27.7 | 16.7/17.4 | 6.5/13.5 | 14.6/16.0 | 12.0/13.0 | **12.3**/14.0 |
| **RLVR on LLaMA3.1-8B-Instruct (Pass@1/Pass@k)** | | | | | | | | |
| Baseline | 0.0/0.0 | 0.1/1.1 | 21.6/41.7 | 3.7/4.6 | 1.5/7.8 | 6.0/17.0 | 1.2/5.0 | 3.8/9.0 |
| + P@1 T. | 0.0/0.0 | 0.1/0.9 | 29.2/38.0 | 12.1/13.2 | 3.8/8.5 | 12.1/14.7 | 5.3/7.4 | 8.7/11.1 |
| + P@k T. | 0.0/0.0 | 0.1/0.9 | 30.5/39.3 | 12.9/14.8 | 5.5/12.2 | 12.2/14.7 | 7.5/10.9 | 9.9/13.0 |
| + P@k T. + P@1 T. | 0.0/0.0 | 0.2/1.1 | 34.4/44.7 | 12.5/14.2 | 7.5/17.8 | 13.2/15.7 | 8.7/10.3 | **10.8/13.7** |

Table 11: The Pass@1/Pass@k performance of Qwen2.5-7b-Instruct trained on different approaches on various Maze sizes. "P@1 T." and "P@k T." denote the Pass@1 Training and Pass@k Training with analytical derivation, respectively. "FS", "BS", and "AD" denote the full sampling, bootstrap sampling, and analytical derivation, respectively. We report the best performance during the training process of each approach.

| | $7 \times 7$ | $9 \times 9$ | $11 \times 11$ | $13 \times 13$ | $15 \times 15$ | $17 \times 17$ | $19 \times 19$ | $21 \times 21$ | Avg. |
|---|---|---|---|---|---|---|---|---|---|
| + P@1 T. | 36.0/38.2 | 32.4/33.0 | 10.6/11.0 | 14.0/14.0 | 8.1/9.0 | 5.0/5.0 | 2.0/2.0 | 3.0/3.0 | 13.9/14.4 |
| + P@k T. w/ FS | 34.6/67.4 | 26.4/47.6 | 13.7/26.0 | 11.0/18.5 | 8.6/17.6 | 3.0/7.6 | 2.2/7.9 | 1.9/5.6 | 12.7/24.7 |
| + P@k T. w/ BS | 45.3/70.6 | 37.8/51.0 | 15.4/27.0 | 12.8/20.7 | 12.3/19.9 | 3.3/8.8 | 4.8/9.0 | 2.3/6.3 | 16.8/26.7 |
| + P@k T. w/ AD | 86.8/98.2 | 94.6/100.0 | 75.2/98.3 | 55.2/84.6 | 39.2/72.0 | 10.5/29.2 | 16.7/28.3 | 3.5/9.7 | **47.7/65.0** |

## F.5 PASS@K TRAINING ON MoE MODEL ON MULTI-MODAL REASONING TASKS

To further evaluate the effectiveness of Pass@k Training, we conducted tests on an in-house model. Unlike the language dense model used in the previous evaluation, this model is a multi-modal MoE model. Table 12 presents the model's test results on MathVision and MMMU. According to the evaluation results, we can observe that the Pass@k performance of the model can be significantly improved, *i.e.,* from 76.4 to 80.0, showing the improvement in the model's exploration ability. After Pass@k Training, the following Pass@1 Training further enhances the Pass@1 scores of the model. These results show the effectiveness of Pass@k Training on different model architectures and different type of downstream tasks.

Table 12: Pass@1/Pass@k Performance of an in-house MoE model trained on different RLVR approaches.

| Pass@1/Pass@k | MathVision | MMMU | Avg. |
|---|---|---|---|
| baseline | 54.6/72.5 | 71.2/80.2 | 62.9/76.4 |
| + Pass@1 Training | 55.3/74.0 | 72.0/83.7 | 63.7/78.9 |
| + Pass@k Training | 53.9/75.6 | 72.0/84.3 | 63.0/80.0 |
| + Pass@k Training + Pass@1 Training | 56.4/76.8 | 72.3/84.5 | **64.4/80.7** |

## G DIFFERENCE BETWEEN PASS@1 AND PASS@K TRAINING

### G.1 ANALYSIS BASED ON ADVANTAGE VALUE CURVES

To analyze why Pass@k Training can help LLMs escape the local optimum, we first visualize the advantage curves of Pass@1 Training and Pass@k Training across responses with different correctness levels, as in GRPO and its variants, the advantage value depends solely on the correctness of the model's response. Furthermore, we observe that during the optimization process, the advantage

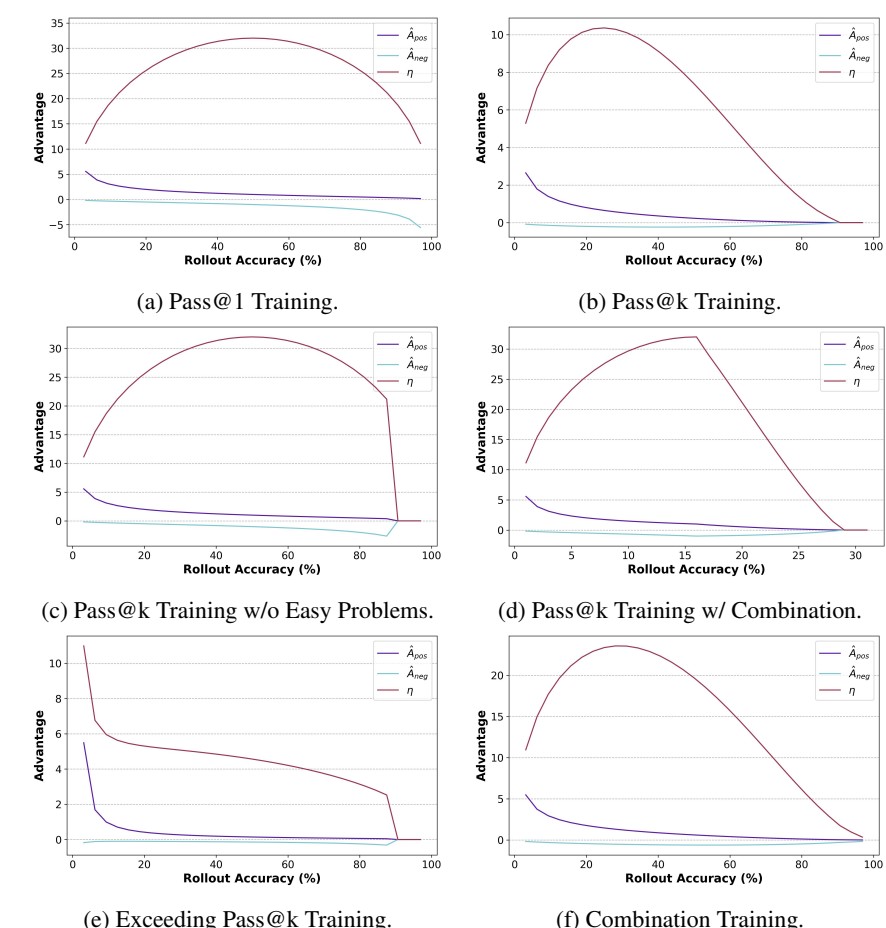

(a) Pass@1 Training.

(b) Pass@k Training.

(c) Pass@k Training w/o Easy Problems.

(d) Pass@k Training w/ Combination.

(e) Exceeding Pass@k Training.

(f) Combination Training.

Figure 14: The curves of the advantage function on the setting of $N_{\text{rollout}} = 32$.

value is directly multiplied by the gradient and can be interpreted as a scaling factor for the gradient. In this context, a larger absolute value of the advantage indicates a greater scaling of the gradient, and thus a larger update step for the corresponding sample. This implies that the model places greater optimization effort on samples with higher advantage magnitudes. Therefore, we argue that the absolute value of the advantage is also an important aspect worthy of investigation. Based on this insight, and to simplify the analysis, we compute the sum of the absolute advantage values across all responses, as defined below:

$$\eta = N_{\text{pos}} \times |\hat{A}_{\text{pos}}| + N_{\text{neg}} \times |\hat{A}_{\text{neg}}|, \tag{21}$$

The curves of $\eta$ (named as *Sum of Absolute Advantage*) are added to our visualization and presented in Fig. 14a and Fig. 14b. Comparing the curves of $\eta$ of Pass@1 Training and Pass@k Training, we can observe that there are three major differences.

**Maximum of Sum of Absolute Advantage $\eta$.** The maximum of $\eta$ of the Pass@1 Training approach is much higher than ones of Pass@k Training approach. As we discussed in Sec. 3.3, the maximum advantage values might affect training efficiency, and adding the coefficient on the loss function to adjust the advantage values can mitigate this issue. Thus, the maximum is not the critical factor that helps Pass@k Training outperform Pass@1 Training.

**Argmax of Sum of Absolute Advantage $\eta$.** According to the curves in Fig. 14a and Fig. 14b, the argmax of $\eta$ are significantly different between Pass@1 and Pass@8 Training. For Pass@1 Training, the maximum of $\eta$ appears at the position of 50% accuracy (*i.e.*, $N_{\text{pos}} = 0.5 \times N_{\text{rollout}}$), while the position of maximum of $\eta$ is 25% accuracy (*i.e.*, $N_{\text{pos}} = 0.25 \times N_{\text{rollout}}$). This phenomenon suggests

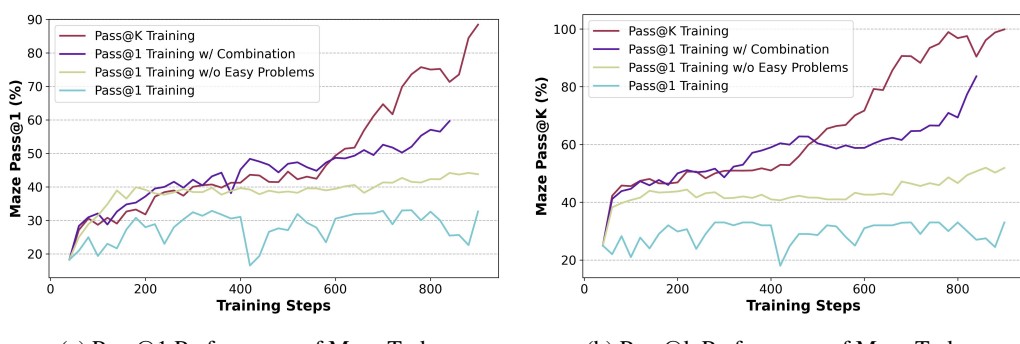

(a) Pass@1 Performance of Maze Tasks.

(b) Pass@k Performance of Maze Tasks.

Figure 15: Training progress of various of Pass@1 Training and Pass@k Training.

that Pass@k Training focuses on optimizing harder problems, while Pass@1 Training focuses on problems with medium difficulty. This further demonstrates that Pass@k Training tends to guide the model toward solving previously unsolved or difficult problems, rather than overfitting to those it has already mastered.

**Trend of Sum of Absolute Advantage** $\eta$. Another key difference between the function curves of Pass@1 and Pass@k Training lies in the trend of the function itself. In the $\eta$ curve of Pass@k Training, the value increases until it reaches a peak, and then gradually decreases to zero. Under this setting, when the problem is relatively easy (*i.e.,* the correctness is higher than 60%), the optimization strength applied by the model (as indicated by the value of $\eta$) becomes smaller than that for harder problems. This further demonstrates that Pass@k Training focuses more on optimizing problems the model has not yet mastered. In contrast, during Pass@1 Training, the $\eta$ curve is symmetric around the point of maximum value, indicating that the training process allocates equal attention to both easy and hard problems.

## G.2 ANALYSIS BASED ON MODEL PERFORMANCE

As we discussed in the previous section, the effectiveness of the argmax and the trend of the sum of absolute advantage $\eta$ still remain unclear. Thus, in this section, we design the corresponding experiments to analyze their effectiveness, based on model performance. Additionally, we designed two training methods that serve as intermediates between Pass@1 and Pass@k Training, *i.e.,* removing the advantage values of the easy problems and combining the advantage estimation approaches of Pass@1 and Pass@k based on the accuracy of the current prompt. The curves of $\hat{A}_{\text{pos}}$, $\hat{A}_{\text{neg}}$, and $\eta$ of these four training approaches are presented in Fig. 14c and Fig. 14d.

First, when the correctness of a response is high, we design the advantage function to decrease gradually toward zero. This setting allows the training reward to increase steadily during the optimization process, indicating that the model avoids getting stuck in a local optimum (*i.e.,* the blue line and purple line). When this optimization is removed, the reward on the training set fails to continue increasing, suggesting that the model has already converged to a local optimum and is no longer learning new knowledge during the RLVR process (*i.e.,* the red line and green line). This phenomenon suggests that excessive learning from easy examples is a key factor causing the model to fall into local optima. Therefore, reducing the degree of learning from easy questions can help prevent the model from getting trapped in such suboptimal solutions.

Second, simply setting the reward for easy questions to zero is not sufficient to effectively prevent the model from over-optimizing on them; it merely delays the point at which the model falls into a local optimum. As shown in Fig. 15, removing the optimization for easy questions (represented by the red line) leads to higher training rewards and better test performance compared to the baseline (represented by the green line). However, both curves exhibit similar trends: after an initial phase of improvement, model performance gradually plateaus, making further progress difficult.

Third, regarding the choice of the argmax position of the $\eta$ function, a comparison of the curves in Fig. 15 reveals that shifting the argmax forward leads to higher optimization efficiency. Specif-

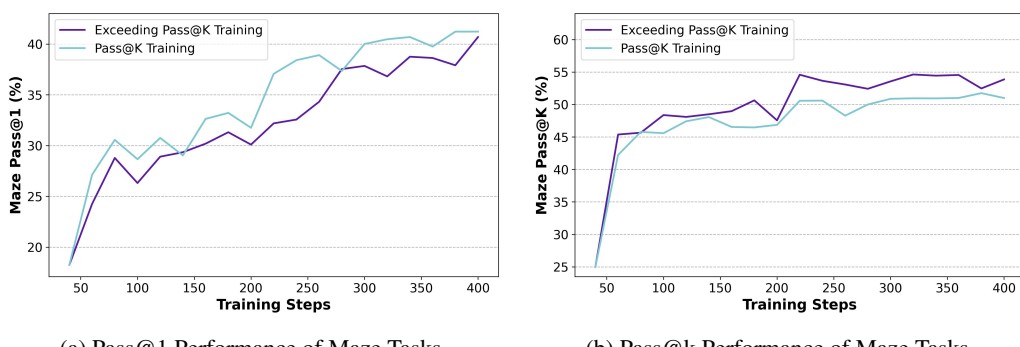

(a) Pass@1 Performance of Maze Tasks.          (b) Pass@k Performance of Maze Tasks.

Figure 16: Training progress of Pass@k Training and Exceeding Pass@k Training.

ically, the model is able to escape from local optima more quickly, and a turning point in training reward appears earlier. This phenomenon suggests that hard problems contribute more significantly to model improvement and yield better optimization effects. Assigning greater optimization strength to harder problems can thus effectively enhance training efficiency, allowing the model to achieve better performance with fewer training steps.

Based on the above results and discussions, several preliminary conclusions can be made, *i.e.,* the argmax of $\eta$ influences the training efficiency and the trend of $\eta$ prevents model from falling into local optimum. Besides, it is important to note that this is only our initial conclusion. More comprehensive experiments tailored to specific tasks and scenarios are required for further validation.

## H    FURTHER ANALYSIS ABOUT PASS@K TRAINING

### H.1    WHAT IS THE GENERALIZATION ABILITY OF LLMS AFTER PASS@K TRAINING?

Table 13: Pass@1/Pass@k Performance of Qwen2.5-7B-Instruct trained on different approaches.

| Pass@1/Pass@k | In-Domain Tasks | | Out-of-Domain Tasks | |
|---|---|---|---|---|
| | ARC-AGI 1 | Enigmate | KORBench | AIME 2025 |
| Qwen2.5-7B-Instruct | 2.4/4.8 | 4.8/10.1 | 36.5/45.9 | 4.2/15.8 |
| + Pass@1 Training | 3.3/3.8 | 12.9/21.3 | 37.7/45.6 | 5.4/19.1 |
| + Pass@k Training | **4.0/5.3** | **17.9/29.8** | **47.7/63.5** | **7.1/22.4** |

To analyze the generalization ability of Pass@k Training, we conduct the corresponding experiments and present the results in Table 13. We can observe that Pass@1 and Pass@k Training can enhance the model's capacities on in-domain and out-of-domain tasks, suggesting the strong generalization ability of the RLVR training process. Further, comparing the performance between these two training approaches, the model trained through Pass@k outperforms the model trained on Pass@1. The reason behind it is that Pass@k Training encourages models to explore better solutions, which can be easily generalized to other tasks. In contrast, Pass@1 Training makes LLMs behave conservatively, thereby affecting LLMs' OOD performance.

**Takeaway from Sec. H.1.** Pass@k Training exhibits stronger generalization ability than Pass@1 Training, achieving greater improvements over the base model in both in-domain and out-of-domain testing.

### H.2    WHETHER THE PASS@K PERFORMANCE CAN BE IMPROVED MORE RAPIDLY?

In previous discussion, we have found that the position of maximum value of $\eta$ will influence the training objective (focus on Pass@1 or Pass@k). Based on these observations and conclusions, we hypothesize that an earlier peak in the $\eta$ function leads to better optimization performance in Pass@k

Training. To test this hypothesis, we design a transformation function as follows:

$$f(N_{\text{pos}}) = \frac{4}{10\log(N_{\text{pos}} + 0.5)}, \quad \hat{A}' = f(N_{\text{pos}}) \times \hat{A}. \tag{22}$$

The advantage value curve after applying the transformation function is shown in Fig. 14e. We observe that, in the transformed curve, the peak of the $\eta$ function is shifted forward to the position where the correctness is $\frac{1}{32}$. According to our hypothesis, such a modification of the advantage function is expected to result in better optimization performance for Pass@k Training. We integrate this transformed function into the RLVR training process (named as *Exceeding Pass@k Training*), and the corresponding training results are presented in Figure 16.

From the experimental results, we observe that Exceeding Pass@k Training can effectively improve the model's Pass@k performance during the early training stage. However, since this method places excessive emphasis on difficult problems, the improvement in Pass@1 performance on downstream tasks progresses more slowly. Based on these observations and analyses, we hypothesize that the computation of advantage values could be adaptively adjusted according to the model's current state. We leave this as a direction for future work.

### H.3 DISCUSSION ABOUT OTHER ENTROPY-GUIDED APPROACHES.

We compare the effectiveness between Pass@k Training and the naive implementation of Entropy-guided Approach (*i.e.,* Entropy Regularization). Moreover, there are several other methods, such as integrating the entropy into the advantage function (Cheng et al., 2025) or focusing on tokens with high covariance (Cui et al., 2025). Similarly, these methods might introduce a new trade-off: overly strict constraints may lead to underfitting and insufficient model training, while overly loose constraints can result in instability during training, potentially affecting the training effectiveness and model performance (He et al., 2025; Hong et al., 2025; Casper et al., 2023), since entropy conflicts with the Pass@1 metric. Therefore, the hyper-parameters should be carefully selected during the above methods to bring the performance improvement of LLMs. Actually, these methods are orthogonal to Pass@k Training, meaning that we can also combine these methods with Pass@k Training to achieve better training results. To verify this, we conduct the experiments in Sec. 4.1 to assess the effectiveness of Pass@k Training based on the guidance of policy entropy, demonstrating significant improvements.

## I USAGE OF LARGE LANGUAGE MODELS IN PAPER WRITING

We only use the large language model to examine and correct the grammar mistakes of our paper.

