# OpenReview forum: "Pass@k Training for Adaptively Balancing Exploration and Exploitation of Large Reasoning Models"
_ICLR.cc/2026/Conference — Submitted to ICLR 2026_

### Official Review · Reviewer_wDyE · 2025-10-25

**Soundness:** 3
**Presentation:** 2
**Contribution:** 2
**Rating:** 6
**Confidence:** 3

**Summary:**

This paper proposes "Pass@k Training" within the RLVR framework for LLM reasoning. The key idea is to use Pass@k metrics (the probability of success within k generated samples) as the optimization objective, rather than the typical Pass@1 focus. The paper introduces three variants of Pass@K training: full sampling, bootstrap sampling, and an analytical derivation method. An extensive empirical investigation explores the effects of Pass@k on model exploration and exploitation, policy entropy, answer diversity, robustness, transferability across domains and model sizes, and synergy with other RLVR strategies. Additional theoretical analysis motivates implicit reward design, suggesting new directions for adaptive RLVR objectives.

**Strengths:**

- This work addresses the Timely and Impactful Problem for improving the performance of RLVR for LLMs.
- The presentation is clear and easy to follow
- Thorough experimental results are presented, including explicit ablation across all variants, comparisons on multiple tasks and domains, diverse LLM architectures/scales, and transfer/generalization studies.
- The paper demonstrated competitive experiment results in RLVR.

**Weaknesses:**

- The paper appears to be largely built upon *"Pass@K Policy Optimization: Solving Harder Reinforcement Learning Problems"*, which introduces PKPO for general RL tasks and provides a more thorough and theoretically grounded analysis. In contrast, this work reads more like an application of PKPO to large language models (LLMs), with limited discussion on what distinguishes it from the original PKPO or what its unique contributions are.

- Both the analytical derivation (AD) in this paper and the analyses in PKPO suggest that Pass@K primarily rescales the advantage function. This rescaling seems equivalent to reducing the penalty for negative responses, as they may still be selected in the positive group. If AD alone is sufficient to characterize the method, it’s unclear why experiments like FS (full sampling) and BS (biased sampling) are necessary, especially since they introduce statistical variability without clear justification.

- I understand that the paper aims to demonstrate the value of Pass@K beyond simply modifying reward signals or adding entropy regularization (as in Section 3.1). However, the strategy of randomly flipping a portion of negative responses to positive seems quite different from Pass@K’s original intent. One could ask: what if we instead directly adjust the reward of negative samples from 0 to some other rewards such as 0.5? I suspect there exists a negative reward setting that could theoretically replicate the effect of Pass@K (which is mentioned as the "implicit reward design?"). Moreover, entropy regularization with a coefficient of 0.001 appears to be a strong baseline. I also suspect that tuning the entropy coefficient around this value could yield competitive results, potentially marginalizing the claimed benefits of Pass@K.

**Questions:**

- The paper states that *“the explicit design of advantage can be regarded as the implicit reward design.”* However, it remains unclear what the implicit reward is actually changed to. While the paper mentions implicit reward design, Sections 4.1 and 4.2 focus primarily on the *“adaptive switch between pass@1 and pass@K.”* It is not clear how this switching mechanism relates to or implements the idea of implicit reward design. A more explicit connection or interpretation would be helpful.

- In Table 3, the analytical derivation (AD) significantly outperforms both FS and BS, despite the fact that these three methods should theoretically behave very similarly. This large gap is surprising and warrants further explanation. What causes AD to perform so much better, and why do FS and BS underperform to such an extent?

- Given that FS, BS, AD and implicit reward design are theoretically expected to be equivalent, a deeper theoretical analysis of their relationships would be more valuable than empirical comparisons among them. Instead, it would be more informative to focus on empirical analyses that offer insight into the actual behavior of Pass@K. For example: 1. A comparison of the advantage function distributions under Pass@1 and Pass@K; 2. A more detailed examination of the probability assigned to top tokens under each method;3. Case studies that directly compare Pass@1 and Pass@K to better understand what Pass@K is effectively doing in practice.

---

> ### Author Response · Authors · 2025-11-17
> **Author Response (Part 1)**
>
> We sincerely thank the reviewer for the insightful suggestions and positive feedback.
>
> **Response to Weaknesses**
>
> > The paper appears to be largely built upon "Pass@K Policy Optimization: Solving Harder Reinforcement Learning Problems", which introduces PKPO for general RL tasks and provides a more thorough and theoretically grounded analysis. In contrast, this work reads more like an application of PKPO to large language models (LLMs), with limited discussion on what distinguishes it from the original PKPO or what its unique contributions are.
>
> Actually, our paper makes unique contributions by proposing valuable insights, including **demonstrating how Pass@k Training balances the exploration and exploitation trade-off of LLMs in the RLVR pipeline**. We directly integrate the Pass@k metric as the reward signal into the widely adopted DAPO framework and conduct comprehensive experiments to dissect its behavior. Our analyses go beyond surface-level observations, highlighting why Pass@k Training matters and how it fundamentally improves the exploitation ability of LLMs (i.e., Pass@1 score).
>
> PKPO and other prior studies [1][2][3] employ Pass@k as the reward function in reinforcement learning and provide rigorous theoretical analyses on reducing variance and bias during training. Building on these foundational works, we investigate how Pass@k Training can be strategically leveraged to boost a model’s Pass@1 score, which is a key metric that directly determines its deployability in real-world applications. Our paper delivers the following significant and actionable insights:
>
> - We show that the **exploration gains introduced by Pass@k Training are not only significant but also fully convertible into exploitation capability**, enabling it to decisively outperform a strong baseline (even the powerful closed-source LLMs) on both Pass@1 and Pass@k metrics (Section 3 and Section G).
>
> - We **extend Pass@k Training to a wide range of scenarios with no need for any complicated derivations**, ensuring it remains straightforward and practical to deploy. This generalization not only simplifies real-world adoption but also delivers additional performance gains on both Pass@1 and Pass@k scores. (Section 4).
>
> - Through experiments and theoretical derivations, we directly **incorporate the Pass@k metric as the reward signal in the widely used RLVR algorithm (i.e., DAPO) in three different approaches**. This establishes a clear and practical pathway for researchers to adopt Pass@k as the reward within current RLVR frameworks. (Section 2).
>
> Building on prior work, we conducted a deeper investigation and analysis, and our contribution is delivering genuinely valuable insights. These insights, which are presented in our paper, provide compelling empirical evidence for the effectiveness of using the Pass@k metric as a reward in RLVR training, most clearly demonstrated by its substantial boost to the Pass@1 metric. Moreover, we integrate this reward into a widely used RLVR algorithm and introduce an implicit reward design strategy, paving the way for its deployment in even more complex and demanding scenarios. Finally, we will revise our paper to present a more detailed discussion.
>
> [1] Pass@K Policy Optimization: Solving Harder Reinforcement Learning Problems
>
> [2] Optimizing Language Models for Inference Time Objectives using Reinforcement Learning
>
> [3] SimKO: Simple Pass@K Policy Optimization

---

> ### Author Response · Authors · 2025-11-17
> **Author Response (Part 2)**
>
> > Both the analytical derivation (AD) in this paper and the analyses in PKPO suggest that Pass@K primarily rescales the advantage function. This rescaling seems equivalent to reducing the penalty for negative responses, as they may still be selected in the positive group. If AD alone is sufficient to characterize the method, it’s unclear why experiments like FS (full sampling) and BS (biased sampling) are necessary, especially since they introduce statistical variability without clear justification.
>
> In our experiments, Full Sampling (FS) and Bootstrap Sampling (BS) can be considered as ablation studies of Analytical Derivation (AD), used to validate the effectiveness of AD and to help readers gradually understand the Pass@k training method.
>
> Additionally, each of these three methods has its own advantages. The FS method can be quickly applied to RLVR algorithms without complex derivations, offering strong adaptability. The BS method also avoids complex derivations while using the Bootstrap Sampling Mechanism to reduce the variance introduced by the sampling process. The AD method requires a certain degree of derivation but minimizes both the variance and bias introduced by the sampling process as much as possible.
>
> In the paper, by introducing these three methods and conducting related experiments, we demonstrate the impact of sampling variance on RLVR performance, while also supporting subsequent analyses of the model’s exploration and exploitation abilities (Section 3.1). Moreover, providing different methods allows researchers to choose the most suitable algorithm based on practical considerations. For instance, in environments with low sampling variance, the Bootstrap Sampling method can be used to reduce the complexity of derivations.
>
> In summary, the experiments with FS and BS are essential. They validate the effectiveness of the AD algorithm, help readers understand the AD method, and provide researchers with more options to adapt to different scenarios.
>
> > I understand that the paper aims to demonstrate the value of Pass@K beyond simply modifying reward signals or adding entropy regularization (as in Section 3.1). However, the strategy of randomly flipping a portion of negative responses to positive seems quite different from Pass@K’s original intent. One could ask: what if we instead directly adjust the reward of negative samples from 0 to some other rewards such as 0.5? I suspect there exists a negative reward setting that could theoretically replicate the effect of Pass@K (which is mentioned as the "implicit reward design?"). Moreover, entropy regularization with a coefficient of 0.001 appears to be a strong baseline. I also suspect that tuning the entropy coefficient around this value could yield competitive results, potentially marginalizing the claimed benefits of Pass@K.
>
> We compared different noise ratios and various coefficients of entropy regularization, and found that Pass@k Training achieves more stable training performance. To further validate the effectiveness of our method, I followed the reviewer’s suggestion and designed the following three baseline approaches:
>
> - Adjust the reward of negative samples from 0 to 0.5.
> - Entropy regularization with a coefficient of 0.0015.
> - Entropy regularization with a coefficient of 0.0005.
>
> The performance of these three baselines and our Pass@k Training is presented in the following Table.
>
> | Methods                          | Pass@k Score of Maze Task |
> |----------------------------------|:---------------------------:|
> | Pass@1 Training                  | 32.0                      |
> | Reward of negative samples = 0.5 | 41.5                      |
> | Reward of negative samples = 1.0 | 32.4                      |
> | **Pass@k Training w/ Full Sampling (Ours)** | **47.6**                      |
>
> | Methods                               | Pass@k Score of Maze Task |
> |---------------------------------------|:---------------------------:|
> | Entropy regularization coef. = 0.0005 | 32.6               |
> | Entropy regularization coef. = 0.001  | 44.8             |
> | Entropy regularization coef. = 0.0015 | 43.9          |
> | **Pass@k Training w/ Full Sampling (Ours)** | **47.6**   |
>
> We observe that even after adjusting the parameters, it is still not possible to achieve results comparable to Pass@k training. Moreover, methods like Noise Reward and Entropy Regularization require extensive hyperparameter tuning, while Pass@k Training needs no adjustment, whose robustness has been proven in Section 3.3. This indicates that Pass@k training is not merely a simple adjustment of the advantage function, as in those baselines, but rather leverages Pass@k to guide the model in enhancing its exploration ability, leading to a more stable and effective RLVR training process.. Additionally, Section 3.3 demonstrates the robustness of Pass@k training. For different values of k, it consistently improves model performance without requiring any training hyperparameter tuning.

---

> ### Author Response · Authors · 2025-11-17
> **Author Response (Part 3)**
>
> **Response to Questions**
>
> > The paper states that “the explicit design of advantage can be regarded as the implicit reward design.” However, it remains unclear what the implicit reward is actually changed to. While the paper mentions implicit reward design, Sections 4.1 and 4.2 focus primarily on the “adaptive switch between pass@1 and pass@K.” It is not clear how this switching mechanism relates to or implements the idea of implicit reward design. A more explicit connection or interpretation would be helpful.
>
> In Full Sampling, rewards are computed by constructing groups, a process we refer to as **explicit reward design**. In contrast, the Analytical Derivation method does not modify the reward function itself but instead directly adjusts the computation of the advantage function, which we consider **implicit reward design**. Therefore, the evolution from Full Sampling to Analytical Derivation represents a transition from explicit reward design to implicit reward design. Implicit reward design can handle scenarios where the reward function is difficult to formulate. By directly designing the advantage function, it avoids this issue and serves as a generalization of Pass@k training. When the reward function is no longer a simple Pass@k but a more complex function, implicit reward design allows for easier construction of the advantage function, enabling effective RLVR training.
>
> In Sections 4.1 and 4.2, we attempt to combine Pass@1 and Pass@k to leverage the strengths of both. In this scenario, **directly designing the reward function can become highly complex**, as it is difficult to formulate a reward that enables adaptive switching between Pass@1 training and Pass@k training. In such cases, where the reward function is hard to design, we achieve the same effect by **directly designing the advantage function, i.e., implicit reward design**. This allows the model to adaptively balance its exploration and exploitation abilities, broadening the applicability of Pass@k training and demonstrating its strong potential for practical use.
>
> Furthermore, in Appendix Section H.2, we attempt to **further improve the model’s Pass@k score**. However, achieving this goal through reward function design is highly challenging. By **shifting the peak of the advantage function forward (Figure 14e)**, we enable a rapid improvement of the Pass@k score during the early stages of training. This experiment further demonstrates that implicit reward design is one of the key approaches for extending Pass@k training to broader applications, thereby enhancing its generalization capability.
>
> > In Table 3, the analytical derivation (AD) significantly outperforms both FS and BS, despite the fact that these three methods should theoretically behave very similarly. This large gap is surprising and warrants further explanation. What causes AD to perform so much better, and why do FS and BS underperform to such an extent?
>
> The differences and advantages of these three algorithms are analyzed in Sections 2.2–2.4. We think that the main factors affecting performance are the variance and bias introduced during the computation of the advantage function.
>
> During Full Sampling, a total of n/k groups are ultimately constructed. Because the number of groups is relatively small, the variance in computing the advantage values increases, leading to training instability. In Bootstrap Sampling, the introduction of the Bootstrap Sampling Mechanism allows for more thorough use of the responses obtained during the rollout process, reducing training variance. Analytical Derivation further reduces this variance. According to Figures 8(b) and 9(b), the three algorithms perform similarly in the early stages of training. As the number of training steps increases, the impact of variance becomes more pronounced, resulting in the observed ordering FS < BS < AD.

---

> ### Author Response · Authors · 2025-11-17
> **Author Response (Part 4)**
>
> > Given that FS, BS, AD and implicit reward design are theoretically expected to be equivalent, a deeper theoretical analysis of their relationships would be more valuable than empirical comparisons among them. Instead, it would be more informative to focus on empirical analyses that offer insight into the actual behavior of Pass@K. For example: 1. A comparison of the advantage function distributions under Pass@1 and Pass@K; 2. A more detailed examination of the probability assigned to top tokens under each method;3. Case studies that directly compare Pass@1 and Pass@K to better understand what Pass@K is effectively doing in practice.
>
> Theoretical and experimental analyses are both crucial components of our work. To better understand the characteristics of FS, BS, AD, and implicit reward design, we conduct analyses from two perspectives: the distribution of advantage values and the model’s behavioral patterns.
>
> **Advantage function distributions**
> To compare the differences between Pass@k Training and Pass@1 Training, in Appendix Section G.1 and G.2, we proposed the theoretical and experimental analyses, respectively.
>
> We visualize the advantage distribution in Figure 14(a) and Figure 14(b), to express the value of the advantage function of different training approaches. We can observe the three major differences:
>
> -  Maximum of the Sum of Absolute Advantage
> - Argmax of the Sum of Absolute Advantage
> - Trend of the Sum of Absolute Advantage
>
> To evaluate the influence of the above three factors, we conduct the experiment and present the results in Figure 15. Based on these results. We can conclude that the Argmax of the Sum of Absolute Advantage influences training efficiency, and the Trend of the Sum of Absolute Advantage prevents the model from falling into a local optimum.
>
> **Case Study of the Model's Behaviours**
>
> We conduct a case study about the generated response of the model trained through Pass@1 Training and Pass@k Training in Section 3.2.
>
> We compare the accuracy and the distribution of different answers among the negative responses from Pass@k and Pass@1 training, to evaluate the exploration ability of LLMs when dealing with uncertain answers. As shown in Figure 4a, the answer diversity within negative responses remains largely unchanged during Pass@1 training, suggesting that the LLM tends to choose “safe” actions and produce similar outputs. This behavior restricts exploration and limits the effectiveness of RLVR. In contrast, Pass@k training encourages the model to achieve a higher Pass@k score, naturally leading it to generate more diverse answers when uncertain. Consequently, the LLM’s exploration ability is enhanced, which in turn strengthens its exploitation capability.

---

### Official Review · Reviewer_nShJ · 2025-10-31

**Soundness:** 2
**Presentation:** 3
**Contribution:** 1
**Rating:** 4
**Confidence:** 4

**Summary:**

Current RLVR training primarily relies on the Pass@1 reward which causes models to prefer conservative and similar actions, suppressing exploration and leading to local optima. To solve this, the paper introduces Pass@k Training, a method that adopts the Pass@k metric as the reward signal. This approach has a higher tolerance for incorrect responses and incentivizes the model to generate diverse solutions to enhance exploration.

**Strengths:**

- This method achieves significant performance improvements with a relatively lightweight modification to the RLVR training process. The gains are not only in  performance but also in sample diversity, as measured by answer diversity and policy entropy .
- The final "Analytical Derivation" method is technically solid, supported by a clear theoretical derivation.
- The experiments are conducted across a wide range of domains, which demonstrates the generalizability and robustness of the method.

**Weaknesses:**

- The novelty of this paper seems to be insufficient. The core motivation and the primary solution appear to be similar with existing research, namely Pass@K Policy Optimization. Besides the effective "annealing" strategy of starting with k>1 and decaying to k=1 is also explored in Pass@K Policy Optimization.
- The implement only adopts GRPO as its training algorithm, the generalizability of the proposed method to other classic RLVR algorithm is not verified.

**Questions:**

- In Table 4, the performance gains are sometimes exaggerated on specific benchmarks with specific base model. For example, in 7B-model experiment Pass@k Training shows a massive improvement on Crypto compared to Pass@1 Training, and P@k T. + P@1 T. shows even much more massive improvement than others. Could the authors provide more analysis on why the benefits of this method are so model-specific and task-specific in these cases?
- What inspired the Pass@k -> Pass@1 transfer strategy? The paper frames this as an exploration-to-exploitation transition, but how did the authors determine the optimal timing or switch-point for this transition.

---

> ### Author Response · Authors · 2025-11-17
> **Author Response (Part 1)**
>
> We sincerely thank the reviewer for the insightful suggestions.
>
> **Response to Weaknesses**
>
> > The novelty of this paper seems to be insufficient. The core motivation and the primary solution appear to be similar with existing research, namely Pass@K Policy Optimization. Besides the effective "annealing" strategy of starting with k>1 and decaying to k=1 is also explored in Pass@K Policy Optimization.
>
> Our paper delivers a meaningful contribution by providing valuable and actionable insights about **how Pass@k Training balances the exploration and exploitation balance of LLMs in RLVR**. Motivated by previous work [1][2][3], we employ the Pass@k metric directly as the reward in the widely used DAPO framework and run extensive experiments to thoroughly probe its dynamics. Our analysis makes it clear why Pass@k Training is essential and how it fundamentally enhances the exploitation capability of LLMs, as reflected by the substantial gains in Pass@1 performance. Actually, the "annealing" strategy in PKPO [1] cannot outperform Pass@1 Training according to Figure 3(a), which fails to demonstrate the practical value of applying the Pass@k metric as the reward.
>
> PKPO and other prior studies [1][2][3] employ Pass@k as the reward function in reinforcement learning and provide rigorous theoretical analyses on reducing variance and bias during training. Building on these foundational works, we ask a more practical and impactful question: **how can Pass@k Training be leveraged to balance the exploration and exploitation ability of LLMs and then boost a model’s Pass@1 score?** This is a key metric for determining real-world deployability. Our paper delivers the following valuable and actionable insights:
>
> - We show that the **exploration gains introduced by Pass@k Training are not only significant but also fully convertible into exploitation capability**, enabling it to decisively outperform a strong baseline (even the powerful closed-source LLMs) on both Pass@1 and Pass@k metrics (Section 3 and Section G).
>
> - We **extend Pass@k Training to a wide range of scenarios with no need for any complicated derivations**, ensuring it remains straightforward and practical to deploy. This generalization not only simplifies real-world adoption but also delivers additional performance gains on both Pass@1 and Pass@k scores. (Section 4).
>
> - Through experiments and theoretical derivations, we directly **incorporate the Pass@k metric as the reward signal in the widely used RLVR algorithm (i.e., DAPO) in three different approaches**. This establishes a clear and practical pathway for researchers to adopt Pass@k as the reward within current RLVR frameworks. (Section 2).
>
> Building on prior work, we conducted a deeper investigation and analysis, and our contribution is delivering genuinely valuable insights. These insights, which are presented in our paper, provide compelling empirical evidence for the effectiveness of using the Pass@k metric as a reward in RLVR training, most clearly demonstrated by its substantial boost to the Pass@1 metric. Moreover, we integrate this reward into a widely used RLVR algorithm and introduce an implicit reward design strategy, paving the way for its deployment in even more complex and demanding scenarios.
>
> As suggested by the reviewer, the previous work (e.g., PKPO) [1][2][3] should be mentioned in the Introduction Section, and we will revise our paper to present a more detailed discussion.
>
> [1] Pass@K Policy Optimization: Solving Harder Reinforcement Learning Problems
>
> [2] Optimizing Language Models for Inference Time Objectives using Reinforcement Learning
>
> [3] SimKO: Simple Pass@K Policy Optimization

---

> ### Author Response · Authors · 2025-11-17
> **Author Response (Part 2)**
>
> > The implement only adopts GRPO as its training algorithm, the generalizability of the proposed method to other classic RLVR algorithm is not verified.
>
> To verify the generalizability of Pass@k Training on other classic RLVR algorithms, we conduct additional experiments in the following table on other classic RLVR algorithms
>
> | Model                                | Crypto | Arithmetic | Logic | Grid | Graph | Search | Sequential | Overall     |
> |--------------------------------------|--------|------------|-------|------|-------|--------|------------|-------------|
> | Qwen2.5-7B-Instruct                  | 0.7    | 3.3        | 48.4  | 9.2  | 11.6  | 1.0    | 5.4        | 10.1        |
> | + ForkingToken [1]                   | 94.0   | 35.0       | 67.1  | 25.9 | 21.6  | 13.4   | 11.7       | 29.1        |
> | **+ ForkingToken [1] & Pass@k Training** | 91.3   | 48.3       | 67.6  | 27.4 | 27.8  | 22.3   | 18.6       | **34.0 (+4.9)** |
> | + GRPO [2]                           | 0.3    | 13.0       | 65.6  | 19.4 | 14.4  | 2.6    | 12.4       | 17.3        |
> | **+ GRPO [2] & Pass@k Training**         | 0.7    | 17.7       | 64.2  | 20.8 | 18.2  | 4.0    | 12.7       | **18.6 (+1.3)** |
> | + DAPO [3]                           | 5.7    | 28.0       | 68.4  | 22.8 | 20.4  | 6.8    | 12.7       | 21.3        |
> | **+ DAPO [3] & Pass@k Training**         | 39.7   | 63.0       | 74.0  | 27.8 | 21.5  | 12.3   | 18.3       | **29.8 (+8.5)** |
>
> We can observe the improvement in Pass@k scores of adopting Pass@k Training on other classic RLVR algorithms, verifying the generalization of our Pass@k Training
>
> [1] Beyond the 80/20 Rule: High-Entropy Minority Tokens Drive Effective Reinforcement Learning for LLM Reasoning
>
> [2] DeepSeekMath: Pushing the Limits of Mathematical Reasoning in Open Language Models
>
> [3] DAPO: An Open-Source LLM Reinforcement Learning System at Scale
>
> **Response to Questions**
>
> > In Table 4, the performance gains are sometimes exaggerated on specific benchmarks with specific base model. For example, in 7B-model experiment Pass@k Training shows a massive improvement on Crypto compared to Pass@1 Training, and P@k T. + P@1 T. shows even much more massive improvement than others. Could the authors provide more analysis on why the benefits of this method are so model-specific and task-specific in these cases?
>
> In our experiment, Pass@k training leads to performance improvements for all models across almost all tasks.
>
> In Table 4, in the Crypto task, the model is required to encrypt plaintext into ciphertext according to specific rules, which is a task where the correct answer can be obtained through simple exploration. Therefore, by enhancing the model’s exploration ability through Pass@k training, significant improvements can be achieved in subsequent Pass@1 training.
>
> Different models exhibit varying levels of exploration ability. Specifically, the 32B model has stronger exploration capabilities than the 7B model. As a result, Pass@k training leads to more noticeable improvements for the 7B model.
>
> In summary, Pass@k training can enhance the exploration ability of all models across nearly all scenarios. The degree of improvement depends on the task difficulty and the model’s inherent capabilities. Difficult tasks require more training to achieve gains, whereas simple tasks can benefit from just a small amount of training.
>
> > What inspired the Pass@k -> Pass@1 transfer strategy? The paper frames this as an exploration-to-exploitation transition, but how did the authors determine the optimal timing or switch-point for this transition.
>
> In our paper, we discuss several strategies for Pass@k → Pass@1 transfer. We found that Pass@k training can adaptively balance exploration and exploitation. Additionally, the entropy metric can help Pass@k training achieve an even better balance between the two abilities.
>
> - In section 2, we can observe that the Pass@k Training can also enhance both the Pass@1 scores and Pass@k of LLMs (Table 3), indicating that the **model’s exploration and exploitation abilities are adaptively balanced without any manual intervention during Pass@k training**。
>
> - In Section 3, we experimented with manually switching the training algorithm to balance exploration and exploitation, achieving promising results (Table 4 and Table 5). We found that this approach requires determining the switch point based on the model’s training curve. To address this issue, we introduce entropy to adaptively balance the two abilities of the model (Section 4).
>
> - To better balance the exploration and exploitation ability of LLMs, in section 4, we introduce **another metric (i.e., Entropy) to help determine the switch-point of the Pass@k -> Pass@1 transfer**. We adopt the Pass@1 Training to improve the exploitation ability when entropy is high, and we employ the Pass@k Training to enhance the model's exploration ability when entropy is low.

---

### Official Review · Reviewer_rZxo · 2025-11-01

**Soundness:** 2
**Presentation:** 2
**Contribution:** 2
**Rating:** 2
**Confidence:** 3

**Summary:**

The paper addresses a key limitation of current RLVR methods, which typically optimize for Pass@1, causing models to converge to conservative local optima rather than fully exploring the reasoning space. To mitigate this, the authors propose Pass@k Training, a framework designed to achieve a more adaptive balance between exploration and exploitation in LRMs.

They first demonstrate that using Pass@k as the reward effectively enhances the model's exploration ability. Then, to improve efficiency and stability, they introduce bootstrap sampling and an analytical derivation for advantage estimation. Furthermore, the paper analyzes why Pass@k Training works and what benefits it brings from four perspectives, and finally extends the idea to an implicit reward design framework that generalizes the analytical formulation to broader RLVR settings.

**Strengths:**

The paper focuses on an important issue in reinforcement learning for large reasoning models—the exploration–exploitation trade-off under verifiable rewards. Its motivation is clear, and the proposed Pass@k Training framework provides a systematic three-stage implementation (full sampling → bootstrap sampling → analytical derivation). The presentation of ablations on different k values and learning rates demonstrates a good empirical sense of robustness.
From a methodological perspective, the analytical form of the group advantage reduces sampling variance and clarifies the relationship between exploration and implicit reward shaping, which can be a useful pedagogical contribution for the RLVR community. The experiments are extensive across multiple reasoning benchmarks and clearly report Pass@1 and Pass@k metrics.

**Weaknesses:**

1. The paper shows strong similarity to Pass@K Policy Optimization: Solving Harder Reinforcement Learning Problems (DeepMind, 2025) in terms of research motivation (limitations of Pass@1 exploration), core idea (training with Pass@k as the reward), key formulation (Eq. 11), and experimental validation (different tasks but similar evaluation goals and performance trends). However, the prior work is only briefly mentioned in the related-work section without any detailed analytical or empirical comparison.

2. The paper never demonstrates that $\mathbb{E}[A_{\text{pos}}]$ or $\mathbb{E}[A_{\text{neg}}]$ are equivalent (or linearly related) to the true gradient of the Pass@k objective $\nabla_\theta \mathbb{E}[\max_i R_i]$. The authors simply substitute group-level reward statistics into the GRPO advantage term, assuming it introduces no bias. In contrast, Pass@K Policy Optimization (PKPO) rigorously proves unbiasedness of its estimator $\hat{\nabla}$ via the policy gradient theorem (Theorems 2–4), establishing theoretical consistency with $\nabla_\theta \text{pass@k}$. This paper skips that derivation entirely and only claims variance reduction without verifying the potential bias.

3. Appendix C defines the group-level variance as $\sigma_{\text{group}} = \sqrt{\bar{R}(1 - \bar{R})}$, which corresponds to a binomial assumption and represents only the *empirical variance* of sampled rewards, not the variance of the estimator itself. The paper does not prove that $\mathrm{Var}[A_{\text{pos}}]$ or $\mathrm{Var}[\nabla_\theta J(\theta)]$ is bounded or convergent as $k$ increases. In contrast, PKPO Section 4 provides formal variance-reduction results using leave-one-out and maxg@(k−1) baselines. Such theoretical guarantees are entirely missing here.

4. The experimental section lacks direct comparisons with related XPO methods such as PKPO and Best-of-N, and does not report variance or significance tests, making it difficult to verify whether the claimed "improvement in exploration capability" and "outperformance over larger models" are statistically reliable.

5. Figure 1 introduces Enigmata scores on the first page without prior explanation of what Enigmata is. The benchmark and its evaluation protocol are only described several pages later, which disrupts the logical flow and makes the early figure hard to interpret for readers unfamiliar with the dataset.

**Questions:**

1. In Section 3 and Appendix C, the derivation of Pass@k training lacks a formal proof of unbiasedness or variance bounds. Could the authors clarify whether the proposed advantage estimator $A_{pos}$ and $A_{neg}$ are theoretically consistent with the true gradient of the Pass@k objective? How does the analysis differ from that of PKPO (Theorem 2–4 in Pass@K Policy Optimization)?

2. Why does the experimental section not include a direct comparison with PKPO or other XPO-style methods (e.g., DAPO, Dr. GRPO)? Since these baselines also optimize preference-based or set-level rewards, such a comparison would clarify whether the proposed method yields distinct benefits or merely reproduces known trends.

---

> ### Author Response · Authors · 2025-11-17
> **Author Response (Part 1)**
>
> We sincerely thank the reviewer for the insightful suggestions.
>
> **Response to Weaknesses**
>
> > The paper shows strong similarity to Pass@K Policy Optimization: Solving Harder Reinforcement Learning Problems (DeepMind, 2025) in terms of research motivation (limitations of Pass@1 exploration), core idea (training with Pass@k as the reward), key formulation (Eq. 11), and experimental validation (different tasks but similar evaluation goals and performance trends). However, the prior work is only briefly mentioned in the related-work section without any detailed analytical or empirical comparison.
>
> Actually, our paper focuses on a more practical and impactful question: **how can Pass@k Training be leveraged to balance the exploration and exploitation ability of LLMs and then boost a model’s Pass@1 score?** Our major contribution is to present valuable insights into it. Concretely, we directly integrate the Pass@k metric as the reward signal into the widely adopted DAPO framework and conduct comprehensive experiments to dissect its behavior. Our analyses go beyond surface-level observations, highlighting why Pass@k Training matters and how it fundamentally improves the exploitation ability of LLMs (i.e., Pass@1 score).
>
> The PKPO and other previous studies [1][2][3] utilize Pass@k as the reward function in reinforcement learning, and propose rigorous theoretical analysis about how to reduce variance and bias during the training process. Inspired by these interesting studies, we wonder how Pass@k Training can be utilized to enhance the model's Pass@1 score, which is one of the critical indicators that determines whether a model can be deployed in real-world applications. Here are the valuable insights our paper proposed:
>
> - We show that the **exploration gains introduced by Pass@k Training are not only significant but also fully convertible into exploitation capability**, enabling it to decisively outperform a strong baseline (even the powerful closed-source LLMs) on both Pass@1 and Pass@k metrics (Section 3 and Section G).
>
> - We **extend Pass@k Training to a wide range of scenarios with no need for any complicated derivations**, ensuring it remains straightforward and practical to deploy. This generalization not only simplifies real-world adoption but also delivers additional performance gains on both Pass@1 and Pass@k scores. (Section 4).
>
> - Through experiments and theoretical derivations, we directly **incorporate the Pass@k metric as the reward signal in the widely used RLVR algorithm (i.e., DAPO) in three different approaches**. This establishes a clear and practical pathway for researchers to adopt Pass@k as the reward within current RLVR frameworks. (Section 2).
>
> Building on prior work, we conducted a deeper investigation and analysis, and our contribution is delivering genuinely valuable insights. These insights, which are presented in our paper, provide compelling empirical evidence for the effectiveness of using the Pass@k metric as a reward in RLVR training, most clearly demonstrated by its substantial boost to the Pass@1 metric. Moreover, we integrate this reward into a widely used RLVR algorithm and introduce an implicit reward design strategy, paving the way for its deployment in even more complex and demanding scenarios.
>
> As suggested by the reviewer, the previous work (e.g., PKPO) [1][2][3] should be mentioned in the Introduction Section, and we will revise our paper to present a more detailed discussion.
>
> [1] Pass@K Policy Optimization: Solving Harder Reinforcement Learning Problems
>
> [2] Optimizing Language Models for Inference Time Objectives using Reinforcement Learning
>
> [3] SimKO: Simple Pass@K Policy Optimization

---

> ### Author Response · Authors · 2025-11-17
> **Author Response (Part 2)**
>
> > - The paper never demonstrates that  or  are equivalent (or linearly related) to the true gradient of the Pass@k objective . The authors simply substitute group-level reward statistics into the GRPO advantage term, assuming it introduces no bias. In contrast, Pass@K Policy Optimization (PKPO) rigorously proves unbiasedness of its estimator  via the policy gradient theorem (Theorems 2–4), establishing theoretical consistency with . This paper skips that derivation entirely and only claims variance reduction without verifying the potential bias.
> >- Appendix C defines the group-level variance as , which corresponds to a binomial assumption and represents only the empirical variance of sampled rewards, not the variance of the estimator itself. The paper does not prove that  or  is bounded or convergent as  increases. In contrast, PKPO Section 4 provides formal variance-reduction results using leave-one-out and maxg@(k−1) baselines. Such theoretical guarantees are entirely missing here.
>
> For these two questions, previous work [1][2] has proposed using the Pass@k metric as a reward function and has already provided rigorous theoretical proofs.
>
> Inspired by these studies, we incorporate the Pass@k metric into the reward function of the widely used RLVR algorithm DAPO, and experimentally demonstrate that Pass@k Training can enhance the model’s exploration ability, and then improve its exploitation capability.
>
> We empirically verify that Pass@k Training outperforms Pass@1 Training, demonstrating from an empirical perspective that Pass@k Training is a more stable and effective training approach. The formal proofs regarding variance and bias are not the focus of this paper， and we plan to build upon prior work to establish these results in future research.
>
> [1] Pass@K Policy Optimization: Solving Harder Reinforcement Learning Problems
>
> [2] Optimizing Language Models for Inference Time Objectives using Reinforcement Learning
>
> > The experimental section lacks direct comparisons with related XPO methods such as PKPO and Best-of-N, and does not report variance or significance tests, making it difficult to verify whether the claimed "improvement in exploration capability" and "outperformance over larger models" are statistically reliable.
>
> In Section 2, we conducted an ablation study and compared common baselines in Figure 3, demonstrating the effectiveness of our method. To further illustrate its efficacy, we organized additional comparison experiments, with the results presented in the table below. From the experimental results, we can see that our method outperforms previous approaches, effectively enhancing the model’s reasoning capabilities.
>
> | Model                    | Crypto      | Arithmetic  | Logic        | Grid        | Graph       | Search      | Sequential  | Overall     |
> |--------------------------|:-------------:|:-------------:|:--------------:|:-------------:|:-------------:|:-------------:|:-------------:|:-------------:|
> | Qwen2.5-7B-Instruct      | 0.1 / 0.7   | 1.0 / 3.3   | 28.1 / 48.4  | 3.7 / 9.2   | 3.0 / 11.6  | 0.3 / 1.0   | 2.7 / 5.4   | 4.7 / 10.1  |
> | + DAPO                   | 1.2 / 5.7   | 6.6 / 28.0  | 41.1 / 68.4  | 14.8 / 22.8 | 14.5 / 20.4 | 3.3 / 6.8   | 9.6 / 12.7  | 12.9 / 21.3 |
> | + ForkingToken [1]           | 78.9 / 94.0 | 10.7 / 35.0 | 46.2 / 67.1  | 19.6 / 25.9 | 14.6 / 21.6 | 7.4 / 13.4  | 10.0 / 11.7 | 20.6 / 29.1 |
> | + GRPO [2]                   | 0.1 / 0.3   | 2.8 / 13.0  |  36.7 / 65.6 | 12.3 / 19.4 | 11.0 / 14.4 | 0.8 / 2.6   | 8.6 / 12.4  | 10.4 / 17.3 |
> | + PKPO [3]                   | 0.0 / 0.0   | 1.6 / 8.0   | 32.4 / 43.6  | 11.5 / 20.5 | 9.8 / 12.7  | 3.7 / 8.1   | 8.9 / 10.2  | 10.1 / 15.4 |
> | + EntropyAdv [4]            | 46.7 / 73.3 | 31.3 / 69.3 | 48.6 / 72.7  | 24.4 / 31.5 | 17.0 / 31.5 | 8.8 / 19.5  | 16.2 / 18.7 | 23.3 / 35.8 |
> | **+ P@k T. + P@1 T. (Ours)** | 96.9 / 98.3 | 36.2 / 67.7 | 49.3 / 71.8  | 30.9 / 37.5 | 20.3 / 30.7 | 25.8 / 37.5 | 10.6 / 12.9 | **30.8 / 40.6** |

---

> ### Author Response · Authors · 2025-11-17
> **Author Response (Part 3)**
>
> > The experimental section does not report variance or significance tests, making it difficult to verify whether the claimed "improvement in exploration capability" and "outperformance over larger models" are statistically reliable.
>
> To make the evaluation results more convincing, we conducted multiple assessments and averaged the outcomes (Section B.2 in Appendix), a practice that is now widely recognized [5][6][7]. To strengthen the paper’s credibility, we followed the reviewers’ suggestion and provided the mean and variance of multiple runs for different algorithms on the Enigmata task. As shown in the table below.
>
> | Pass@1 Score             | mean@8 | std@8 |
> |--------------------------|:--------:|:-------:|
> | GPT-4o-1120              | 14.2   | -     |
> | + P@1 T.                 | 12.9   | 1.91  |
> | + P@k T.                 | 17.9   | 4.70  |
> | **+ P@k T. + P@1 T. (Ours)** | **30.8**   | **7.44**  |
>
> | Pass@k Score             | mean@8 | std@8 |
> |--------------------------|:--------:|:-------:|
> | + P@1 T.                 | 21.3   | 0.36  |
> | + P@k T.                 | 29.8   | 0.73  |
> | **+ P@k T. + P@1 T. (Ours)** | **40.6**   | **0.28**  |
>
> By comparing the performance of “P@1 T.” and “P@k T.” on the Pass@k score, we find that the difference in their means is more than three times the variance, indicating with 99.7% confidence that Pass@k training achieves higher Pass@k scores than Pass@1 training. Similarly, by comparing “+P@1 T.” and “+P@k T. + P@1 T.” on the Pass@1 score, we find that the difference in their means exceeds twice the variance, providing 95.4% confidence that our method achieves better Pass@1 scores than the traditional Pass@1 training approach.
>
> Similarly, by comparing the performance of “GPT-4o-1120” and “+P@k T. + P@1 T.”, we find that the difference in their average Pass@1 score also exceeds twice the variance, demonstrating that our method enables the 7B model to outperform larger models.
>
> [1] Beyond the 80/20 Rule: High-Entropy Minority Tokens Drive Effective Reinforcement Learning for LLM
>
> [2] DeepSeekMath: Pushing the Limits of Mathematical Reasoning in Open Language Models
>
> [3] Pass@K Policy Optimization: Solving Harder Reinforcement Learning Problems
>
> [4] Reasoning with Exploration: An Entropy Perspective
>
> [5] DeepSeek-R1: Incentivizing Reasoning Capability in LLMs via Reinforcement Learning
>
> [6] DAPO: An Open-Source LLM Reinforcement Learning System at Scale
>
> [7] Qwen3 Technical Report
>
> > Figure 1 introduces Enigmata scores on the first page without prior explanation of what Enigmata is. The benchmark and its evaluation protocol are only described several pages later, which disrupts the logical flow and makes the early figure hard to interpret for readers unfamiliar with the dataset.
>
> We thank the reviewer's suggestions. Due to space constraints, we briefly introduced our experimental setup and benchmarks at the beginning of Section 2, referring readers to the relevant appendix sections to help them better understand our method. In future revisions, we will provide more detailed descriptions in the appropriate sections to give readers a clearer understanding of the downstream tasks.
>
> **Response to Questions**
> > In Section 3 and Appendix C, the derivation of Pass@k training lacks a formal proof of unbiasedness or variance bounds. Could the authors clarify whether the proposed advantage estimator  and  are theoretically consistent with the true gradient of the Pass@k objective? How does the analysis differ from that of PKPO (Theorem 2–4 in Pass@K Policy Optimization)?
>
> For these questions, prior work [1][2] has already proposed using the Pass@k metric as a reward function and provided rigorous theoretical guarantees. Building on these studies, we integrate the Pass@k metric into the reward function of the widely adopted RLVR algorithm DAPO and experimentally show that Pass@k Training not only strengthens the model’s exploration ability but also translates these gains into improved exploitation.
>
> In our work, we want to answer a more practical and impactful question: how can Pass@k Training be leveraged to balance the exploration and exploitation ability of LLMs and then boost a model’s Pass@1 score? Our empirical results clearly demonstrate that Pass@k Training outperforms Pass@1 Training, establishing it as a more stable and effective approach. While formal proofs regarding variance and bias are beyond the scope of this paper, we intend to extend these theoretical foundations in future work.
>
> [1] Pass@K Policy Optimization: Solving Harder Reinforcement Learning Problems
>
> [2] Optimizing Language Models for Inference Time Objectives using Reinforcement Learning

---

> ### Author Response · Authors · 2025-11-17
> **Author Response (Part 4)**
>
> > Why does the experimental section not include a direct comparison with PKPO or other XPO-style methods (e.g., DAPO, Dr. GRPO)? Since these baselines also optimize preference-based or set-level rewards, such a comparison would clarify whether the proposed method yields distinct benefits or merely reproduces known trends.
>
> Our method is compatible with other XPO-style approaches. In our paper, we have already demonstrated that it can be successfully applied to DAPO. To further verify the generalization capability of Pass@k training, we conducted additional experiments, including ForkingToken [1], GRPO [2], and DAPO [3]. The Pass@k scores of the Engimata task are shown in the table below.
>
> | Model                                | Crypto | Arithmetic | Logic | Grid | Graph | Search | Sequential | Overall     |
> |--------------------------------------|--------|------------|-------|------|-------|--------|------------|-------------|
> | Qwen2.5-7B-Instruct                  | 0.7    | 3.3        | 48.4  | 9.2  | 11.6  | 1.0    | 5.4        | 10.1        |
> | + ForkingToken [1]                   | 94.0   | 35.0       | 67.1  | 25.9 | 21.6  | 13.4   | 11.7       | 29.1        |
> | **+ ForkingToken [1] & Pass@k Training** | 91.3   | 48.3       | 67.6  | 27.4 | 27.8  | 22.3   | 18.6       | **34.0 (+4.9)** |
> | + GRPO [2]                           | 0.3    | 13.0       | 65.6  | 19.4 | 14.4  | 2.6    | 12.4       | 17.3        |
> | **+ GRPO [2] & Pass@k Training**         | 0.7    | 17.7       | 64.2  | 20.8 | 18.2  | 4.0    | 12.7       | **18.6 (+1.3)** |
> | + DAPO [3]                           | 5.7    | 28.0       | 68.4  | 22.8 | 20.4  | 6.8    | 12.7       | 21.3        |
> | **+ DAPO [3] & Pass@k Training**         | 39.7   | 63.0       | 74.0  | 27.8 | 21.5  | 12.3   | 18.3       | **29.8 (+8.5)** |
>
> From the results above, we find that our method can be effectively adapted to XPO-style approaches while achieving improvements in both Pass@1 and Pass@k scores.
>
> Additionally, for other baseline approaches, we compare the model performance curves during training in the paper (Figure 3). To further validate the effectiveness of our method, we compare Pass@k training with PKPO [4] and EntropyAdv [5], and present the results (Pass@1 scores) in the table below.
>
> | Model                    | Crypto      | Arithmetic  | Logic        | Grid        | Graph       | Search      | Sequential  | Overall     |
> |--------------------------|:-------------:|:-------------:|:--------------:|:-------------:|:-------------:|:-------------:|:-------------:|:-------------:|
> | Qwen2.5-7B-Instruct      | 0.1 / 0.7   | 1.0 / 3.3   | 28.1 / 48.4  | 3.7 / 9.2   | 3.0 / 11.6  | 0.3 / 1.0   | 2.7 / 5.4   | 4.7 / 10.1  |
> | + DAPO                   | 1.2 / 5.7   | 6.6 / 28.0  | 41.1 / 68.4  | 14.8 / 22.8 | 14.5 / 20.4 | 3.3 / 6.8   | 9.6 / 12.7  | 12.9 / 21.3 |
> | + ForkingToken           | 78.9 / 94.0 | 10.7 / 35.0 | 46.2 / 67.1  | 19.6 / 25.9 | 14.6 / 21.6 | 7.4 / 13.4  | 10.0 / 11.7 | 20.6 / 29.1 |
> | + GRPO                   | 0.1 / 0.3   | 2.8 / 13.0  |  36.7 / 65.6 | 12.3 / 19.4 | 11.0 / 14.4 | 0.8 / 2.6   | 8.6 / 12.4  | 10.4 / 17.3 |
> | + PKPO                   | 0.0 / 0.0   | 1.6 / 8.0   | 32.4 / 43.6  | 11.5 / 20.5 | 9.8 / 12.7  | 3.7 / 8.1   | 8.9 / 10.2  | 10.1 / 15.4 |
> | + EntropyAdv            | 46.7 / 73.3 | 31.3 / 69.3 | 48.6 / 72.7  | 24.4 / 31.5 | 17.0 / 31.5 | 8.8 / 19.5  | 16.2 / 18.7 | 23.3 / 35.8 |
> | **+ P@k T. + P@1 T. (Ours)** | 96.9 / 98.3 | 36.2 / 67.7 | 49.3 / 71.8  | 30.9 / 37.5 | 20.3 / 30.7 | 25.8 / 37.5 | 10.6 / 12.9 | **30.8 / 40.6** |
>
> [1] Beyond the 80/20 Rule: High-Entropy Minority Tokens Drive Effective Reinforcement Learning for LLM
>
> [2] DeepSeekMath: Pushing the Limits of Mathematical Reasoning in Open Language Models
>
> [3] DAPO: An Open-Source LLM Reinforcement Learning System at Scale
>
> [4] Pass@K Policy Optimization: Solving Harder Reinforcement Learning Problems
>
> [5] Reasoning with Exploration: An Entropy Perspective

---

### Official Review · Reviewer_A5FH · 2025-11-12

**Soundness:** 2
**Presentation:** 3
**Contribution:** 2
**Rating:** 4
**Confidence:** 3

**Summary:**

The paper proposes Pass@k Training for large language models using reinforcement learning with verifiable rewards (RLVR) to improve exploration. It argues that existing RLVR methods use Pass@1 as the reward metric, leading to conservative policies that prefer safe, similar actions and limit exploration. The authors adopt Pass@k, a metric that measures whether correct responses appear within k attempts, as the reward signal to encourage diverse solution generation. The paper presents three implementations: full sampling, bootstrap sampling for efficiency, and an analytical derivation that provides closed-form advantage estimates. Experiments are presented across maze navigation, mathematical reasoning (AIME, OlymMATH), and synthetic puzzles (Enigmata). Results show that Pass@k Training improves Pass@k performance while maintaining or improving Pass@1 scores. The paper further introduces implicit reward design, suggesting that combining the advantage values of Pass@1 and Pass@k Training, and an adaptive policy entropy guidance based Pass@k Training may be more effective than Pass@k Training.

**Strengths:**

Originality

-  The analytical derivation of advantage estimates (Equations 8-9) is an original technical contribution, providing closed-form solutions that eliminate sampling variance and reduce computational overhead compared to full sampling.

Quality

- Experiments have been presented across diverse domains, including maze navigation with varying sizes, mathematical reasoning tasks, and synthetic puzzles.
- The paper presents diversity metrics and policy entropy measurements showing Pass@k Training maintains high entropy and generates diverse negative responses while Pass@1 Training causes entropy collapse.

Clarity

- The paper is generally well-written with clear motivation and effective visualizations (Figure 2 comparing training procedures).

Significance

-  Pass@k Training results in a 7B model achieving 30.8% overall Pass@1 accuracy on Enigmata after Pass@k followed by Pass@1 training, surpassing the performance of GPT-4o and approaching that of Claude 3.7, and improvements transferring across model scales (7B, 32B) and task domains.

**Weaknesses:**

1. The connection between the advantage function design presented in Section 4 and optimality guarantees is not established.
2. All experiments use outcome-based verification with binary rewards, and applicability to process-based rewards, partial credit, or continuous reward signals remains unexplored.
3. The computational cost analysis is incomplete. Actual wall-clock time comparisons across all methods and scales are not provided.  Depending on k, Pass@k may significantly increase the wall-clock time, making training impractical for certain tasks.
4. The comparison with baseline methods is limited, missing recent RLVR approaches beyond noise rewards and entropy regularization, and there is no comparison with other exploration methods, such as curiosity-driven learning or count-based bonuses.
5. The hyperparameter selection for k and learning rates appears task-specific without any proposed principled guidance to select them.
6. The reported results do not report the number of runs/seeds, variance, or significance tests, making the claims weak.

Minor

7. The implicit reward design section (Section 4) feels somewhat disconnected from the main narrative and could be better integrated or expanded into a separate contribution.

**Questions:**

1. Can the authors provide formal convergence guarantees for Pass@k Training? Under what conditions does the method converge to optimal or near-optimal policies, and what is the sample complexity compared to Pass@1 Training?
2. How does Pass@k Training perform with non-binary rewards or process-based verification where intermediate steps receive partial credit? Would the analytical derivation extend to these settings?
3. What is the actual wall-clock time and computational cost (GPU hours, memory usage) for all three variants (full sampling, bootstrap sampling, analytical derivation) across different model scales and tasks?
4. Can the authors provide principled guidance/heuristics for selecting k and learning rates based on task characteristics? For instance, do task difficulty, solution space size, or reward sparsity predict optimal hyperparameter settings?

---

> ### Author Response · Authors · 2025-11-17
> **Author Response (Part 1)**
>
> We sincerely thank the reviewer for the insightful suggestions.
>
> **Response to Weaknesses**
>
> >The connection between the advantage function design presented in Section 4 and optimality guarantees is not established.
>
> For the unbiased Pass@k metric (i.e., Eq. 3), its optimal solution is the same as that of the Pass@1 metric. Ideally, if all outputs generated by the model are positive, both Pass@k and Pass@1 achieve their optimal values. When the model may generate incorrect outputs, neither Pass@k nor Pass@1 can reach 1, i.e., the optimal solution cannot be attained. Therefore, we can consider that the optimal solutions of Pass@k training and Pass@1 training are the same, and the difference lies only in the paths they take to guide the model’s optimization.
>
> The two application methods mentioned in Section 4 combine Pass@k training and Pass@1 training by using entropy as an indicator to adaptively choose between Pass@1 or Pass@k training. In this scenario, since both have the same optimal solution under ideal conditions, the optimal solution after combining them is also the same as that of Pass@1 training, i.e., all outputs generated by the model are positive.Based on our theoretical analysis, we also conducted related experiments in Section 4 and found that our proposed advantage function design method can better optimize the model, achieving superior results compared to Pass@1 training.
>
> > All experiments use outcome-based verification with binary rewards, and applicability to process-based rewards, partial credit, or continuous reward signals remains unexplored.
>
> In current RLVR-related work [1][2][3], researchers mainly focus on using only outcome-based verification as a supervision signal to guide the training of large language models.
>
> Following the setup of this line of work, we explore the impact of Pass@k as a reward function on the model's exploration and exploitation capabilities. Therefore, in our work, we mainly focus on the experimental setup of outcome-based verification. For other setups, e.g., process-based rewards, we will conduct further research in future work.
>
> [1] DeepSeek-R1: Incentivizing Reasoning Capability in LLMs via Reinforcement Learning
>
> [2] Kimi k1.5: Scaling Reinforcement Learning with LLMs
>
> [3] DAPO: An Open-Source LLM Reinforcement Learning System at Scale
>
> > The computational cost analysis is incomplete. Actual wall-clock time comparisons across all methods and scales are not provided. Depending on k, Pass@k may significantly increase the wall-clock time, making training impractical for certain tasks.
>
> First, in our experiment settings, the number of rollouts is set to 32, i.e., $N_\\text{rollout}=32$. In this case, the overhead of each method during the RLVR training process is similar. Second, with the increase of the value of k, the wall-clock time of Pass@k Training w/ Analytical Derivation will not increase accordingly. This is because the changes in the value of k will not affect the rollout times, thereby Pass@k Training and Pass@1 Training share the same rollout times and require similar wall-clock time for the training process.
>
> The above is the theoretical analysis of the time overhead of Pass@k Training and Pass@1 Training. Below, we provide the actual training time consumed by the model.
>
> | Methods                                      | Training Time | Final Response Length |
> |:----------------------------------------------|:---------------:|:-----------------------:|
> | Pass@1 Training                              | 8h 29min 5s   | 113                   |
> | Pass@k Training w/ FS $N_\\text{rollout}=32$ | 7h 57min 13s  | 117                   |
> | Pass@k Training w/ FS $N_\\text{rollout}=128$ | 26h 5min 19s  | 369                   |
> | Pass@k Training w/ BS                        | 9h 26min 44s  | 394                   |
> | Pass@k Training w/ AD                        | 8h 56min 37s  | 297                   |
>
> Based on the above training time, we can find that the time overhead of Pass@1 Training and Pass@k Training during the training process is of the same order of magnitude. The reason why Pass@k Training takes slightly longer is that, under the training of this method, the length of the responses generated by the model is longer, leading to longer training time. This phenomenon is independent of the value of k, indicating that the value of k in Pass@k Training will not result in a significant increase in training time. Instead, **the number of rollouts affects the training time, but this is not an issue introduced by our method.**

---

> ### Author Response · Authors · 2025-11-17
> **Author Response (Part 2)**
>
> > The comparison with baseline methods is limited, missing recent RLVR approaches beyond noise rewards and entropy regularization, and there is no comparison with other exploration methods, such as curiosity-driven learning or count-based bonuses.
>
> Recently, methods for enhancing a model’s exploration ability have mainly relied on  designing reward functions that encourage exploratory behavior [1] or on using entropy as a metric to measure exploration [2]. In Section 3.1, we compare our method with these two approaches in terms of performance differences. To provide a more comprehensive comparison, we summarize various existing methods for improving exploration ability (i.e., PKPO [1], EntropyAdv [2]), along with RL baselines (i.e., GRPO[3], ForkingToken [4]), and present the comparison results (Pass@1 scores / Pass@k scores) of Engimata tasks in the table below.
>
> | Model                    | Crypto      | Arithmetic  | Logic        | Grid        | Graph       | Search      | Sequential  | Overall     |
> |--------------------------|:-------------:|:-------------:|:--------------:|:-------------:|:-------------:|:-------------:|:-------------:|:-------------:|
> | Qwen2.5-7B-Instruct      | 0.1 / 0.7   | 1.0 / 3.3   | 28.1 / 48.4  | 3.7 / 9.2   | 3.0 / 11.6  | 0.3 / 1.0   | 2.7 / 5.4   | 4.7 / 10.1  |
> | + DAPO                   | 1.2 / 5.7   | 6.6 / 28.0  | 41.1 / 68.4  | 14.8 / 22.8 | 14.5 / 20.4 | 3.3 / 6.8   | 9.6 / 12.7  | 12.9 / 21.3 |
> | + ForkingToken           | 78.9 / 94.0 | 10.7 / 35.0 | 46.2 / 67.1  | 19.6 / 25.9 | 14.6 / 21.6 | 7.4 / 13.4  | 10.0 / 11.7 | 20.6 / 29.1 |
> | + GRPO                   | 0.1 / 0.3   | 2.8 / 13.0  |  36.7 / 65.6 | 12.3 / 19.4 | 11.0 / 14.4 | 0.8 / 2.6   | 8.6 / 12.4  | 10.4 / 17.3 |
> | + PKPO                   | 0.0 / 0.0   | 1.6 / 8.0   | 32.4 / 43.6  | 11.5 / 20.5 | 9.8 / 12.7  | 3.7 / 8.1   | 8.9 / 10.2  | 10.1 / 15.4 |
> | + EntropyAdv             | 46.7 / 73.3 | 31.3 / 69.3 | 48.6 / 72.7  | 24.4 / 31.5 | 17.0 / 31.5 | 8.8 / 19.5  | 16.2 / 18.7 | 23.3 / 35.8 |
> | **+ P@k T. + P@1 T. (Ours)** | 96.9 / 98.3 | 36.2 / 67.7 | 49.3 / 71.8  | 30.9 / 37.5 | 20.3 / 30.7 | 25.8 / 37.5 | 10.6 / 12.9 | **30.8 / 40.6** |
>
> As we discussed in Section H.3 in the Appendix, the entropy-based approach tends to cause instability during training, which negatively impacts performance. In our experiments above, our method outperforms these approaches, demonstrating its superiority.
>
> [1] Pass@K Policy Optimization: Solving Harder Reinforcement Learning Problems
>
> [2] Reasoning with Exploration: An Entropy Perspective
>
> [3] DeepSeekMath: Pushing the Limits of Mathematical Reasoning in Open Language Models
>
> [4] Beyond the 80/20 Rule: High-Entropy Minority Tokens Drive Effective Reinforcement Learning for LLM Reasoning
>
> > The hyperparameter selection for k and learning rates appears task-specific without any proposed principled guidance to select them.
>
> Actually, in Section B.2 in the Appendix, we present the hyperparameters of our experiments. During the training process, we set k as 4 and the learning rate as 1e-6 for all tasks. The selection of the hyperparameters follows the experience proposed by previous work [1][2]. Besides, we evaluate the influence of the value of k for the Pass@k Training process in Section 3.3. We can observe that the different values of k would not largely affect the effectiveness of the training process, showing the robustness of our Pass@k Training.
>
> [1] An Empirical Study on Eliciting and Improving R1-like Reasoning Models
>
> [2] DAPO: An Open-Source LLM Reinforcement Learning System at Scale

---

> ### Author Response · Authors · 2025-11-17
> **Author Response (Part 3)**
>
> > The reported results do not report the number of runs/seeds, variance, or significance tests, making the claims weak.
>
> To make the evaluation results more convincing, we conducted multiple assessments and averaged the outcomes (Section B.2 in Appendix), a practice that is now widely recognized [5][6][7]. The number of runs is 8 and the seed is 42.
>
> To strengthen the paper’s credibility, we followed the reviewers’ suggestion and provided the mean and variance of multiple runs for different algorithms on the Enigmata task. As shown in the table below.
>
> | Pass@1 Score             | mean@8 | std@8 |
> |--------------------------|:--------:|:-------:|
> | GPT-4o-1120              | 14.2   | -     |
> | + P@1 T.                 | 12.9   | 1.91  |
> | + P@k T.                 | 17.9   | 4.70  |
> | **+ P@k T. + P@1 T. (Ours)** | **30.8**   | **7.44**  |
>
> | Pass@k Score             | mean@8 | std@8 |
> |--------------------------|:--------:|:-------:|
> | + P@1 T.                 | 21.3   | 0.36  |
> | + P@k T.                 | 29.8   | 0.73  |
> | **+ P@k T. + P@1 T. (Ours)** | **40.6**   | **0.28**  |
>
> By comparing the performance of “P@1 T.” and “P@k T.” on the Pass@k metric, we observe that the difference in their means is more than three times the variance, indicating 99.7% confidence that Pass@k training achieves higher Pass@k scores than Pass@1 training. Likewise, when comparing “+P@1 T.” and “+P@k T. + P@1 T.” on the Pass@1 metric, the difference in their means exceeds twice the variance, providing 95.4% confidence that our method yields better Pass@1 performance than the conventional Pass@1 training approach. Furthermore, when comparing “GPT-4o-1120” with “+P@k T. + P@1 T.”, we find that the difference in their average Pass@1 score also surpasses twice the variance, demonstrating that our method enables the 7B model to outperform much larger models.
>
> > The implicit reward design section (Section 4) feels somewhat disconnected from the main narrative and could be better integrated or expanded into a separate contribution.
>
> In fact, the implicit reward design mentioned in Section 4 is **a way to extend Pass@k training to broader scenarios**.
>
> In Section 2, the Full Sampling approach implements Pass@k training by designing a reward function, while the Analytical Derivation method achieves it by designing an advantage function. This insight inspires us that directly designing the advantage function (implicit reward design) can avoid complex derivations. In scenarios where the reward function is difficult to design, implicit reward design can achieve the desired objectives without requiring complicated derivations. This approach makes Pass@k training more flexible and enhances its potential for application in real-world and complex scenarios.
>
> In Section 4, we design a scenario where entropy is used to indicate the model’s exploration ability. In this setting, directly designing a reward function is highly challenging, as it is difficult to combine the Pass@k metric and the entropy metric effectively. To enable the application of Pass@k training in this scenario, we employ implicit reward design to seamlessly integrate Pass@k with entropy, thereby achieving the intended training objective, i.e., improving both the Pass@1 score and the Pass@k score.
>
> In summary, Section 4 is closely connected to the preceding sections. Section 2 introduces Pass@k training, Section 3 explores the relationship between Pass@k training and the model’s exploration ability, and Section 4 extends Pass@k training to broader application scenarios, showing a valuable direction for future research.
>
> **Response to Questions**
> >Can the authors provide formal convergence guarantees for Pass@k Training? Under what conditions does the method converge to optimal or near-optimal policies, and what is the sample complexity compared to Pass@1 Training?
>
> Regarding the theoretical justification, previous work [1][2] has introduced the use of the Pass@k metric as a reward function and provided corresponding proofs. Building upon this foundation, our work takes an experimental perspective by applying the Pass@k reward to a popular RLVR algorithm. We further propose novel insights, demonstrating that Pass@k Training not only enhances the model’s exploration ability but also improves its exploitation capability, adaptively balancing the two. Extensive experiments (Section 3 and Section F) validate that Pass@k Training enables the model to approach the optimal policies more effectively than Pass@1 Training.
>
> [1] Pass@K Policy Optimization: Solving Harder Reinforcement Learning Problems
>
> [2] Optimizing Language Models for Inference Time Objectives using Reinforcement Learning

---

> ### Author Response · Authors · 2025-11-17
> **Author Response (Part 4)**
>
> > How does Pass@k Training perform with non-binary rewards or process-based verification where intermediate steps receive partial credit? Would the analytical derivation extend to these settings?
>
> In existing research related to Reinforcement Learning for Visual Reasoning (RLVR) [1][2][3], scholars primarily rely on outcome-based verification as the sole supervisory signal to guide the training of large language models (LLMs). Building upon the experimental framework established by this line of work, this study investigates the influence of the Pass@k metric, when adopted as a reward function, on the model's exploration and exploitation capabilities. Consequently, the core focus of our research lies in the experimental design of outcome-based verification. For alternative setups, such as process-oriented reward mechanisms, we reserve in-depth exploration for future investigations.
>
> For the mentioned scenarios, we believe the Analytical Derivation can also be extended. In Section 4, we propose the implicit reward design, whose purpose is to extend Pass@k Training to more scenarios. For example, in multi-objective RLVR (i.e., non-binary reward scenario), Pass@k Training can be combined with other metrics (e.g., Entropy) through implicit reward design to achieve better training results.
>
> [1] DeepSeek-R1: Incentivizing Reasoning Capability in LLMs via Reinforcement Learning
>
> [2] Kimi k1.5: Scaling Reinforcement Learning with LLMs
>
> [3] DAPO: An Open-Source LLM Reinforcement Learning System at Scale
>
> > What is the actual wall-clock time and computational cost (GPU hours, memory usage) for all three variants (full sampling, bootstrap sampling, analytical derivation) across different model scales and tasks?
>
> The table below shows the training time of Full Sampling, Bootstrap Sampling, and Analytical Derivation on the  Maze task of Qwen2.5-7B-Instruct. From the results, we can see that the choice of k in Pass@k training does not affect training efficiency; only the number of rollouts in the RLVR framework impacts efficiency. Notably, the rollout count is introduced by the RLVR algorithm itself, not by our method.
>
> | Methods                                      | Training Time | Final Response Length |
> |:----------------------------------------------|:---------------:|:-----------------------:|
> | Pass@1 Training                              | 8h 29min 5s   | 113                   |
> | Pass@k Training w/ FS $N_\\text{rollout}=32$ | 7h 57min 13s  | 117                   |
> | Pass@k Training w/ FS $N_\\text{rollout}=128$ | 26h 5min 19s  | 369                   |
> | Pass@k Training w/ BS                        | 9h 26min 44s  | 394                   |
> | Pass@k Training w/ AD                        | 8h 56min 37s  | 297                   |
>
> > Can the authors provide principled guidance/heuristics for selecting k and learning rates based on task characteristics? For instance, do task difficulty, solution space size, or reward sparsity predict optimal hyperparameter settings?
>
> In Section 3.1, we analyze the effects of different k values and learning rates on model performance. We find that, regardless of the choice of k, the model’s exploration ability is consistently enhanced, demonstrating strong robustness. However, when k is set to a larger value, optimization efficiency may decrease — a problem that can be effectively mitigated by increasing the learning rate.
>
> In our experiments (the experiment settings mentioned in Section B.2 in the Appendix), following the conclusions of previous work [1][2], we used a fixed setting of k = 4 and learning rate = 1e-6 for all tasks, without any hyperparameter tuning. This demonstrates that the chosen configuration is effective and that our Pass@k training method exhibits strong robustness — achieving excellent performance with simple hyperparameter choices.
>
> [1] An Empirical Study on Eliciting and Improving R1-like Reasoning Models
>
> [2] DAPO: An Open-Source LLM Reinforcement Learning System at Scale

---

### Author Response · Authors · 2025-11-20
**Update the Revision of Our Paper**

We sincerely thank the reviewer for the insightful suggestions and positive feedback. Following the reviewers’ suggestions, we revised our paper and uploaded a new version. The revisions in the paper have been marked in blue font and include the following contents:
+ We further discussed the previous work and clarified the contributions of our paper: exploring how Pass@k Training balances a model’s exploration and exploitation abilities, as well as its practical value and potential for real-world applications (Abstract & Section 1).
+ We added experiments of Pass@k Training based on other RLVR algorithms and presented the corresponding results, demonstrating that Pass@k Training can effectively enhance model capabilities across different methods. (Section 3.4)
+ We added a comparison between Pass@k Training and the baseline methods to demonstrate that it can effectively balance exploration and exploitation in RLVR. (Section 3.5)
+ We further explained the motivation and significance of implicit reward design. By extending Pass@k Training to various scenarios without complex analytical derivation, implicit reward design makes a significant contribution to its practical application. (Section 4)
+ We presented the mean and variance of the experimental results and conducted validity tests, confirming the effectiveness of Pass@k Training. (Appendix F.1)

---

### Author Response · Authors · 2025-11-26
**Summary of Author Response**

We sincerely thank the reviewer for the insightful suggestions. We present our responses and hope these responses can address the concerns of the reviewers. Now, we summarize our responses in the following.

> The contribution of our work.

PKPO and other prior studies [1][2][3] employ Pass@k as the reward function in reinforcement learning and offer rigorous theoretical analyses on reducing variance and bias during training.

However, these works leave open a far more consequential and practice-oriented question: how can Pass@k Training be harnessed to fundamentally reshape the exploration–exploitation balance of LLMs and ultimately deliver substantial gains in Pass@1 performance, which is the metric that directly determines whether a model can be deployed in real-world applications?

To address this critical gap, we (i) derive how Pass@k Training can be integrated into popular RLVR frameworks (Section 2), (ii) systematically characterize its core properties (Section 3), and (iii) generalize Pass@k Training to a broad set of practical scenarios (Section 4).

Our findings clearly demonstrate that Pass@k Training delivers tangible, consistent, and practically significant improvements, enabling it to move beyond theoretical interest and become a truly deployable training paradigm. This constitutes a highly valuable and impactful contribution to the field.

[1] Pass@K Policy Optimization: Solving Harder Reinforcement Learning Problems

[2] Optimizing Language Models for Inference Time Objectives using Reinforcement Learning

[3] SimKO: Simple Pass@K Policy Optimization

> The adaption of Pass@k Training to other RLVR algorithms.

We attempt to integrate Pass@k Training with multiple reinforcement learning algorithms, and our experiments verify that Pass@k Training consistently enhances the effectiveness of all these algorithms.

We revise our paper, and present the related experiment results and discussion in Section 3.4.

> The comparision between Pass@k Training and other baselines.

We compare the performance of Pass@k Training with other popular baselines, observing that Pass@k Training outperforms these baseline methods.

We revise our paper, and present the related experiment results and discussion in Section 3.5.

> The significance tests of the experiment results

We conduct the significance tests of our experiment results, verifying the effectiveness of Pass@k Training

We revise our paper, and present the related experiment results and discussion in Appendix F.1.

---

### Meta-Review · Area_Chair_ryQE · 2026-01-06

**Summary:**

This paper studies RLVR for large reasoning models and addresses a practically relevant question of what happens if we train using Pass@k instead of the standard Pass@1 objective.
The authors propose Pass@k training, describe three concrete implementations (full sampling, bootstrap sampling, and an analytical advantage formulation), and argue that using Pass@k encourages exploration without harming, and possibly even improving, Pass@1 performance. The paper also introduces an implicit reward design perspective that reframes the analytical advantage construction as a way to interpolate between exploration and exploitation. The empirical evaluation spans maze tasks, synthetic puzzles, and mathematical reasoning benchmarks, with comparisons across several RLVR backbones.

The reviewers mostly agree that the motivation is timely, and that the paper contains a good amount of experimental work. There are also some concerns among the reviewers which is elaborated below.

**Reviewer Concerns:**

There are some concerns across reviews. The main concern appears to be lack of novelty. The novelty relative to existing Pass@k based policy optimization work does not appear to be sufficiently clear. Second, the theoretical grounding of the proposed estimator is weaker than the framing suggests, particularly in comparison to prior work that explicitly proves unbiasedness and variance properties.
These concerns appear to persist even after reading the author responses and the revised manuscript, although the rebuttal does address some of the issues raised by reviewers.

**Reviewer Scores:**

The authors provided detailed and thoughtful responses. They clarified their intended scope, added experiments on additional RLVR backbones, included variance of the experimental results , expanded the discussion of prior work, and improved the presentation of the implicit reward design idea, etc. These changes address several of the more practical and presentation oriented criticisms. However, the central issues around novelty and the absence of new theoretical justification remain largely unchanged. It is a decent paper but may not meet the standard of ICLR by a small margin.

---

### Decision · Program_Chairs · 2026-01-26

Reject